# Deregulated Metabolic Pathways in Ovarian Cancer: Cause and Consequence

**DOI:** 10.3390/metabo13040560

**Published:** 2023-04-15

**Authors:** Roopak Murali, Vaishnavi Balasubramaniam, Satish Srinivas, Sandhya Sundaram, Ganesh Venkatraman, Sudha Warrier, Arun Dharmarajan, Rajesh Kumar Gandhirajan

**Affiliations:** 1Department of Human Genetics, Faculty of Biomedical Sciences Technology and Research, Sri Ramachandra Institute of Higher Education and Research (Deemed to be University), Porur, Chennai 600116, India; 2Department of Radiation Oncology, Sri Ramachandra Medical College & Research Institute, Sri Ramachandra Institute of Higher Education & Research (Deemed to be University), Porur, Chennai 600116, India; 3Department of Pathology, Sri Ramachandra Medical College & Research Institute, Sri Ramachandra Institute of Higher Education & Research (Deemed to be University), Porur, Chennai 600116, India; 4Division of Cancer Stem Cells and Cardiovascular Regeneration, School of Regenerative Medicine, Manipal Academy of Higher Education (MAHE), Bangalore 560065, India; 5Cuor Stem Cellutions Pvt Ltd., Manipal Institute of Regenerative Medicine, Manipal Academy of Higher Education (MAHE), Bangalore 560065, India; 6Department of Biomedical Sciences, Faculty of Biomedical Sciences Technology and Research, Sri Ramachandra Institute of Higher Education and Research (Deemed to be University), Porur, Chennai 600116, India; 7Stem Cell and Cancer Biology Laboratory, Curtin University, Perth, WA 6102, Australia; 8School of Pharmacy and Biomedical Sciences, Curtin University, Perth, WA 6102, Australia; 9Curtin Health and Innovation Research Institute, Curtin University, Perth, WA 6102, Australia

**Keywords:** ovarian cancer, tumor metabolism, oncogenes, tumor suppressor genes

## Abstract

Ovarian cancers are tumors that originate from the different cells of the ovary and account for almost 4% of all the cancers in women globally. More than 30 types of tumors have been identified based on the cellular origins. Epithelial ovarian cancer (EOC) is the most common and lethal type of ovarian cancer which can be further divided into high-grade serous, low-grade serous, endometrioid, clear cell, and mucinous carcinoma. Ovarian carcinogenesis has been long attributed to endometriosis which is a chronic inflammation of the reproductive tract leading to progressive accumulation of mutations. Due to the advent of multi-omics datasets, the consequences of somatic mutations and their role in altered tumor metabolism has been well elucidated. Several oncogenes and tumor suppressor genes have been implicated in the progression of ovarian cancer. In this review, we highlight the genetic alterations undergone by the key oncogenes and tumor suppressor genes responsible for the development of ovarian cancer. We also summarize the role of these oncogenes and tumor suppressor genes and their association with a deregulated network of fatty acid, glycolysis, tricarboxylic acid and amino acid metabolism in ovarian cancers. Identification of genomic and metabolic circuits will be useful in clinical stratification of patients with complex etiologies and in identifying drug targets for personalized therapies against cancer.

## 1. Introduction

Ovarian cancers originate from different cells of the ovary and are a lethal gynecological malignancy with a high mortality rate in women. There are more than 30 types of ovarian tumors based on cellular origin, but the main four types of ovarian tumors include epithelial tumors that develop from the epithelial cells of the ovary, tumors that originate from the ova, tumors that develop from the stromal cells and the extremely rare small cell carcinoma of the ovary (SCCO). They are further classified based on the tissue origin, such as epithelial (7seventypes), germ cell (three types) and stromal cells (two types). Epithelial ovarian cancer (EOC) and its subtypes, high-grade serous cancer (HGSOC) and low-grade serous cancer (LGSOC), dominate the histological types of ovarian cancer (Figure 1). Ovarian cancer accounts for 4% of all cancers in women [1]. According to the American cancer society (2022), women with localized epithelial, stromal and germ cell ovarian carcinomas have a survival rate ranging from 95% to 98%, women with regional metastasized ovarian carcinoma have a survival rate of 75% to 94%, and women with distant metastasized ovarian carcinoma have a survival rate from 31% to 74%. The main symptoms of ovarian cancer include abdominal swelling, abdominal blotting, pelvic pain, and weight loss. Even though there are multiple symptoms of ovarian cancer, this cancer often remains undetected since most of the symptoms overlap with those of abdominal or gastrointestinal diseases and is hence diagnosed in the late stages, which accounts for its lethality [2].

Several theories have been proposed for ovarian carcinogenesis. Perpetual ovulation and frequent ovulatory cycles with follicles containing a high amount of estrogen lead to inflammation, DNA damage, and eventual premalignant transformation of surrounding tissues. Epidemiological studies indicate that women with fewer ovulatory cycles due to breast feeding, pregnancies, and use of oral contraceptives have a reduced risk of ovarian cancer [3,4]. Chronic inflammation of fallopian tubes due to retrograde menstruation, infection, or inflammatory agents also lead to ovarian cancer [5]. Several factors associated with endometriosis lead to chronic inflammatory cytokine release, resulting in malignant transformation [6]. Finally, germline mutations in Breast Cancer Gene 1 (*BRCA1*), Breast Cancer Gene 2 (*BRCA2*), BRCA1 interacting Protein 1 (*BRP1*), Partner and Localizer of BRCA2 (*PALB2*), RAD51 homolog C (*RAD51C*), RAD51 homolog D (*RAD51D*), and individuals with Lynch syndrome are susceptible to ovarian cancers [7,8,9]. Initially, it was believed that ovarian carcinomas are derived from the ovarian surface epithelium and following the sequential events of metaplasia further lead to the development of the various cell types such as serous, mucinous, endometrioid, clear cell, and borderline tumors which constitutes the morphological subtypes of epithelial ovarian carcinomas. Recently, due to the progression and advancements of histopathological, molecular, and genetic studies, a better model was developed for understanding ovarian carcinogenesis. This dualistic model displays two broad categories, which are designated as type I, where precursor lesions in the ovary have been clearly described, and type II, where there is no clear description of such lesions and these tumors may originate de novo from the fallopian tube and/or the surface epithelium of the ovary [10,11]. Type I tumors consist of three sub-groups: (i) endometriosis-related tumors, which includes endometrioid, clear cell, and seromucinous carcinomas; (ii) low-grade serous carcinomas; (iii) mucinous carcinomas and malignant Brenner tumors. Type II tumors mainly include high-grade serous carcinomas that can be further classified based on morphologic and molecular characteristics. Type II tumors are highly aggressive in nature and they are present in advanced stage in >75% of the cases, whereas type I tumors are present during the early stages and they are less aggressive. Type I tumors fairly respond to chemotherapy, and type II tumors respond very well to the chemotherapeutic drugs but they have a high chance of recurrence. Type I tumors are characterized by infrequent *TP53* mutations, whereas frequent rates of *TP53* mutations have been observed in type II tumors [12]. Both tumor types are characterized by the somatic and/or germline mutations in different genes. For example, in LGSOC, mutations have been observed in Kirsten rat sarcoma viral oncogene homolog (*KRAS*) and v-raf murine sarcoma viral oncogene homolog B1 (*BRAF*). Inactivating mutations of AT-rich interactive domain-containing protein 1A (*ARID1A*) and Phosphatase and TENsin homolog deleted on chromosome 10 (*PTEN*); activating mutations of catenin β1 gene (*CTNNB1*), phosphatidylinositol-4,5-bisphosphate 3-kinase catalytic subunit alpha (*PIK3CA*) have been observed in endometrioid carcinomas [11,13,14,15,16]. In HGSOC, the most observed mutations were of *TP53*. Other than TP53, somatic and germline mutations of *BRCA1/2* and mutations in neurogenic locus notch homolog protein 3 (*NOTCH3*) and cyclin E1 (*CCNE1*) were also a contributing factor of HGSOC carcinogenesis [17,18]. Current treatment modalities for ovarian cancer include surgery, chemotherapy, and targeted therapy and limited radiotherapy. Given the complexity of the disease and a wide range of pathological manifestations, there is a lacuna in identification of prognostic biomarkers and targeted drugs in ovarian cancer. Despite the presence of well-known biomarkers such as VEGF, PARP, EGFR, and folate receptors that are targeted against by drugs that act as their inhibitors in ovarian cancer treatment, there are several obstructions that challenge the refinement for their clinical application using targeted therapies. This can be because of the sophisticated molecular mechanisms of the disease that are yet to be understood. Because of this, even though a range of predictive biomarkers have been identified, none have proven to be potent and reliable [19]. In this review, we return to the basics and discuss the current understanding of acquired genetic mutations during ovarian carcinogenesis and its direct role in metabolic dependencies of the tumor to further strengthen the scope of therapeutics targeting tumor metabolism in ovarian cancers.

### 1.1. Somatic Driver Mutations

Like most human malignancies, ovarian carcinoma occurs due to the accumulation of mutations in the genes that regulate growth and control cell division and proliferation. Two of the most important classes of genes that regulate cell growth and proliferation are oncogenes and tumor suppressor genes. Activation of an oncogene together with an inactivation of a tumor suppressor gene leads to carcinogenesis. Figure 2 represents the overall survival rate of the ovarian cancer patients who are having mutations in the major oncogenes and tumor suppressor genes using online tool Kaplan–Meier plotter [20]. Furthermore, there is progressive accumulation of additional somatic driver mutations that contribute to clonal evolution of the tumor, leading to metabolic adaptations and epithelial to mesenchymal transition. Table 1 lists the genes that have been implicated in the pathogenesis and progression of ovarian cancers. The frequency of somatic and germline mutations was obtained from the cBioportal web server [21].

#### 1.1.1. *Her-2/neu*

The *Her-2/neu* gene is a human homolog of neu gene. This is a widely expressed gene that is located in chromosome 17 (17q21-22). The *Her-2/neu* codes for c-erbB 2, which is a 185 kDa transmembrane structured growth factor receptor with tyrosine kinase activity [44]. Activation of the protein coded by this gene in turn leads to activation of the phosphorylation cascade of the PI3K-Akt pathway, which is a key regulator of cell proliferation [45]. The *Her-2/neu* gene was overexpressed in different types of cancer, including ovarian cancer. In a recent study conducted in 98 patients who were diagnosed with EOC, 22.4% were reported to have HER2 overexpression [23]. In another study, immunohistochemistry (IHC) and in situ hybridization (ISH) were performed in tissue microarrays of ovarian clear cell carcinoma (OCCC), endometrial clear cell carcinoma, and mixed endometrial carcinoma. Using IHC when *HER2* expression was observed, there was a difference in expression of HER2 in ovarian clear cell carcinoma and endometrial clear cell carcinoma and there was a marked discordance in ISH [22]. In one study, the expression of HER2 was correlated with the expression of Kinesin Family member 2A (*KIF2A*), a gene that codes for kinesin-like protein and plays a key role in normal spindle activity during mitosis. IHC was performed in 111 ovarian cancer and fallopian cancer as well as 48 control groups for *HER2*. *KIF2A* mRNA levels were measured using RT-PCR in 15 ovarian cancer and 20 control groups. *KIF2A* mRNA levels were overexpressed and both HER2 and KIF2A overexpression was seen in IHC. No correlation between *KIF2A* and *HER2* expression was discovered. The patients with overexpression of both showed poor survival rate [24].

#### 1.1.2. *c-MYC*

*c-MYC* is an important oncogene that codes for a transcription factor, c-Myc, that plays a crucial role in cell proliferation. *c-MYC* is located on 8q24 [46]. *c-MYC* codes for 143 amino acid rich DNA binding protein with a dimerization domain that consists of a helix–loop–helix leucine zipper motif in the C-terminal end [47]. Deregulation of *c-MYC* was a triggering factor for oncogenesis in cancers, especially in ovarian cancers. In one of the first studies conducted to determine the role of *c-MYC* in ovarian cancer, this gene was amplified in ovarian cancer cell lines [27]. In one study, fluorescent in situ hybridization (FISH) was performed on 507 tumor micro-arrays of ovarian cancer. A 38.5% increase in the copy number of the *c-MYC* was discovered to play a role in development of ovarian cancer [25]. In a recent study conducted using 64 early staged epithelial ovarian cancer tissues and 20 normal ovarian tissues, the expression of two genes—*c-MYC* and Plasmacytoma Variant Translocation 1 (*PVT1*)—was studied. There was a significantly higher copy number variation for *c-MYC* and *PVT1* in EOC tissues than in the normal ovarian tissues. There was also increased expression of *c-MYC* and *PVT1* in the EOC tissues compared to the control groups [26].

#### 1.1.3. *KRAS*

Kirsten rat sarcoma virus (*KRAS*), one of the three members of the RAS family, is one of the most common oncogenes that are associated with oncogenesis. *KRAS* is located on 12p12.1 [48] and codes for 21 kDa proteins. Ras proteins are believed to display GTPase activity and are involved in a cascade of serine/threonine kinases such as Mitogen-Activated Protein Kinase Kinase (MAPKK) that plays an important role in cell differentiation [49,50]. When *KRAS* is mutated and inactivated, it leads to improper and autonomous differentiation that triggers malignancy. Mutated *KRAS* were involved in the development of ovarian cancer. A study was carried out on 71 epithelial ovarian carcinomas, which comprised of mucinous and non-mucinous subtypes. *KRAS* mutation was identified in codon 12 in eight mucinous carcinomas and codon 13 in one mucinous carcinoma. In two non-mucinous carcinomas, codon 12 alteration was seen in the serous carcinoma and codon 13 alteration was seen in the endometrioid carcinoma [30]. In another study, out of 95 mucinous ovarian tumors, 63% of the ovarian tumors showed codon 12 mutations, whereas 11.5% of ovarian tumors showed codon 13 mutations. Eight tumors exhibited both codon 12 and 13 mutations [29]. In a further study, codon 12 mutations for *KRAS* were studied. A total of 381 malignant and 22 benign ovarian cancer tissues were analyzed using a biochip platform. Mutations for *KRAS* were identified in 15% of the samples, and in those samples, codon 12 mutations were identified in 92% of the samples [28].

#### 1.1.4. *BTAK*

*BTAK* is a gene that codes for Aurora-A, which is a serine/threonine kinase that plays an important role in the promotion and progression of mitosis. The *BTAK* is located on chromosome 20q13. *BTAK* is believed to be amplified and overexpressed in cancers and especially in breast cancer [51]. Amplification and overexpression of *BTAK* has also been observed in ovarian cancer. In one study, knock down or silencing the expression of Aurora-A kinase led to decreased tumor growth. This study also showed there is an inverse correlation of Aurora-A expression with *BRCA2* expression. Knockout of Aurora-A promoted the expression of *BRCA2* and other tumor suppressor genes [52]. In another study, protein expression and kinase activity of Aurora-A kinase was analyzed in 92 primary ovarian tumor samples. Kinase activity was observed in 48% of the samples and protein overexpression was seen in almost 57% of the samples. High protein expression was correlated with increased kinase activity. The expression was observed even in low-grade non-invasive tumors [31]. A study conducted by Landen et al. also showed overexpression of Aurora-A kinase protein in EOC. Out of 70 patient samples, 82% showed Aurora-A kinase overexpression [32].

#### 1.1.5. *TP53*

Tumor protein p53 (*TP53*) is one of the most important tumor suppressor genes that codes for p53 protein which is a chief transcription factor that regulates various promoters playing a crucial role in regulating the expression of wide range of genes. *TP53* is located at 17p13.1 and consists of 11 exons. This gene codes for a 53 kDa p53 protein that has four functional domains: one transcriptional activation, one tetramerization and two DNA binding domains. *TP53* has been extensively studied for its role in regulating key cellular processes that are involved in controlling cell proliferation and its role in maintaining the integrity and stability of the genome [53,54]. *TP53* is believed to be mutated in almost every type of cancer. It undergoes different types of mutations such as loss of function (LOF), gain of function (GOF) and loss of heterozygosity (LOH) that exposes the cells to oncogenic change by depriving the cells of anticancer protection [55]. LOH mutations have been frequently observed in ovarian cancer. A study conducted by G Saretzki et al. showed that in invasive ovarian carcinomas, there was a frequency of 56% for LOH at a locus near 17p13 (*TP53*). This study also showed invasive ovarian carcinomas have a high frequency of LOH compared to the non-invasive types [35]. TP53 LOH plays a role in conferring cisplatin resistance in ovarian cancer. A study conducted by Plisiecka-Hałasa et al. suggests that LOH in the TP53 loci 17p13, a dysfunctional mutation, confers cisplatin resistance in ovarian cancer patients [56]. Similarly, polymorphism K351N mutation in TP53 loci causes cisplatin resistance in EOC. Patients with K351N mutation showed recurrence of ovarian cancer within 6 months in spite of receiving cisplatin chemotherapy [57]. Musaffe Tuna et al. studied several mutational hotspots in TP53 locus and identified Y163C mutation in hotspot group which leads to loss of function of TP53. This results in compromised AKT/Notch pathway leading to cisplatin resistance and poor prognosis in ovarian cancer cells [58]. In another study carried out by comparing low-grade and high-grade serous ovarian carcinoma, infrequent *TP53* mutations in low-grade carcinomas were shown, whereas the mutations were ubiquitous in high-grade serous ovarian carcinoma, in which 100% *TP53* mutations were demonstrated [33]. Another study showed the role of missense mutations in serous ovarian carcinoma. This study showed that 66% of ovarian cancers exhibited missense mutations of *TP53*. This study also showed early-stage cancers had a significantly higher rate of missense mutations than the advanced stage of this disease [34].

#### 1.1.6. *BRCA1*

*BRCA1* is a major tumor suppressor gene that is mainly expressed in the breast and ovaries. The *BRCAl* is located at 17q21-12, and this gene consists of 22 coding exons. The *BRCA*1 codes for a 220 kDa protein [59,60]. *BRCA1* has been shown to play an important role in transcription of RNA since it was associated with RNA polymerase II holoenzyme [61]. *BRCA1* has been commonly detected mutated in breast cancers, but studies have shown *BRCA1* mutations have also been associated with ovarian cancers. *BRCA1* mutations are frequently found in hereditary ovarian cancers. A case study conducted in women of Ashkenazi Jewish ethnicity with inherited mutations for *BRCA1* showed that 54% of women who had *BRCA1* mutations had a lifetime risk for developing ovarian cancer [36]. In a study conducted by Stratton et al., 374 women with ovarian cancer were studied for germ line *BRCA1* mutations. A total of 13 women had germ line *BRCA1* mutations. Out of the 13 mutations, 12 mutants expressed a truncated protein product. Out of these 12 women who expressed truncated protein, 9 had a family history of breast and ovarian cancer [37]. A secondary intragenic mutation of *BRCA1* tends to restore the wild type version of this allele. Elizabeth et al. showed that in women with ovarian cancer who had the *BRCA1* mutation and who were resistant to platinum, a secondary mutation of *BRCA1* restored the wild-type reading frame [38].

#### 1.1.7. *BRCA2*

*BRCA2* is another important tumor suppressor gene that is also associated with hereditary ovarian cancer. *BRCA2* is located at 13q12-1. The 70 kb genomic DNA consists of 26 coding exons and this gene codes for a 400 kDa protein product [40,62]. Unlike *BRCA1* which is expressed highly in the breast, *BRCA2* is expressed at higher levels in the testis and thymus but expressed in lower levels in the breast and ovary [63]. Since *BRCA2* plays a role in hereditary ovarian cancer, several studies have been conducted on ways in which *BRCA2* is involved in ovarian cancer. An earlier study has shown that *BRCA2* does not pose a major threat in the development of ovarian cancer [40]. In a population-based study conducted by Zhang et al., out of 1342 women diagnosed with ovarian cancer, 67 women had mutations for *BRCA2*. This study also showed that age and ethnicity of the women was associated with higher prevalence for both *BRCA1* and *BRCA2* mutations [41]. Similar to *BRCA1*, a secondary mutation in *BRCA2* has also been shown to restore the mutant *BRCA2* reading frame into the wild-type reading frame in cisplatin-resistant ovarian cancer [39].

#### 1.1.8. *PTEN*

Phosphatase and tensin homologue (*PTEN*) is an important tumor suppressor gene known for its role in the inhibition of the phosphoinositide-3 kinase (PI3K) pathway. *PTEN* is located at chromosome 10q23.31. It encodes the 403-amino acid protein that possesses both lipid and protein phosphatase activities [64]. *PTEN* mutations were key players in ovarian carcinogenesis. Driver mutations of *PTEN* have been identified in endometrioid and clear cell subtypes of ovarian cancer [42,43]. Homozygous loss of *PTEN* was observed in 6% of HGSOC [17]. A combination of *PTEN* mutations with *KRAS* mutations was determined to induce highly invasive and metastatic endometrioid ovarian cancer [65].

## 2. Deregulated Metabolic Pathways in Ovarian Cancer

Cancer cells thrive on cellular metabolism to facilitate their growth, uncontrolled proliferation, invasiveness, and metastasis. In cancer cells, multiple metabolic pathways were altered compared to their normal counterparts so that these cancer cells survive and sustain themselves against the changing conditions in the tumor microenvironment. Even though altered metabolism is an important hallmark of cancer mentioned by Hanahan and Weinberg, it is one of the most understudied hallmarks of cancer [66]. In the next section, we summarize important metabolic pathways that were deregulated in ovarian cancer. Table 2 lists the proteins that are differentially expressed in major metabolic pathways in ovarian cancer.

### 2.1. Glycolysis

Glucose is an important molecule that plays a central role in the energy generation since the oxidation of glucose carbon dioxide and water results in a standard free-energy change of −2840 kJ/mol. Glycolysis is considered an important metabolic pathway that breaks down glucose into two three-carbon compounds. During this process, free energy is released in the form of high-energy deriving molecules such as adenosine triphosphate (ATP) and reduced form of Nicotinamide Adenine Dinucleotide (NADPH). The process of glycolysis is a series of enzyme-catalyzed reactions to yield two molecules of pyruvate. Glycolysis is initiated when glucose is phosphorylated at the hydroxyl group of C-6. It consists of two phases: (i) preparatory phase where two molecules of ATP are invested, and (ii) payoff phase where energy is gained. Glycolysis occurs in the cell cytoplasm under anaerobic conditions. Cancer cells undergo a modified form of glycolysis called aerobic glycolysis or the Warburg effect in which the cells rapidly proliferate and there is an increased glucose uptake and lactate production even in the presence of oxygen [67]. Figure 3 shows the differentially expressed proteins in the glycolysis pathway, represented as boxplots obtained from GEPIA web server uploaded by Zhang et al., 2017 [68]. Almost 60% of the ATP is generated by tumor cells in the presence of aerobic conditions through glycolysis [69]. The Warburg effect has been shown to play a role in progression of ovarian cancer. A study conducted by Teng et al. showed silencing of *AKT2* (AKT serine/threonine kinase 2) and *AKT3* (AKT serine/threonine kinase 3), isoforms of AKT which are a downstream mediator of PI3K signaling pathway, was determined to regulate the Warburg effect in EOC [70].

Hepatocyte nuclear factor 1β (HNF1β) is a transcription factor involved in the development of kidney and pancreatic beta cells. Overexpression of HNF1β was associated with altered glucose metabolism in OCCC by promoting increased glucose uptake and increased aerobic glycolysis [71]. The tumor growth in cancer is associated with metabolic reprogramming of nitric oxide (NO), especially in cancers such as ovarian cancer. Caneba et al. showed that in ovarian cancer, NO is involved in regulating tumor growth and inhibits mitochondrial respiration, shifting these cells towards glycolysis so that production of ATP is maintained. NO was also discovered to decrease reactive oxygen species (ROS) levels by increasing the levels of NADPH and glutathione [72]. Glucose transporter 1 (GLUT1), which is the first component of glycolysis, is an isoform of glucose transporters that plays an important role in transporting glucose into cells. *GLUT1* was overexpressed in different tumors and especially in ovarian cancer. In addition, the increased expression of GLUT1 was associated with poor survival rate in ovarian cancer patients [73]. Ovarian cancer was reported to rely upon the GLUT1 transporter to regulate glycolysis and tumor growth. A study showed that silencing GLUT1 expression blocks stress-regulated glycolysis and anchorage-dependent and independent growth of ovarian cancer cells [74]. Forkhead box protein M1 (FOXM1), a transcription factor, was associated with glycolysis. *FOXM1*, *GLUT1* and hexokinase 2 (*HK2*) was upregulated in EOC. Molecular investigations show that FOXM1 binds directly to the *GLUT1* and *HK2* promoter regions and regulates the expression of the genes at the transcriptional level. Knockdown of FOXM1 significantly reduced the expression of *GLUT1* and *HK2* genes and also downregulated aerobic glycolysis as well as cell proliferation [75]. Targeting *GLUT1* by microRNA (miR)-144 and silencing it exhibited a metabolic shift in glucose uptake, proving that miR-144 can regulate GLUT1 expression and aerobic glycolysis [76]. Hexokinase 2 (HK2) is an important enzyme that catalyzes the conversion of glucose to glucose-6-phosphate, which is one of the initial steps in glycolysis. HK2 was also shown to have a critical role in ovarian carcinogenesis. HK2 was overexpressed in ovarian cancer. The expression of HK2 is associated with advanced stage and high-grade cancers. HK2 was also discovered to regulate lactate production and was also linked with cancer metastasis [77]. Upregulation of miR-603 was reported to diminish the malignant behavior of ovarian cancer cells by targeting aerobic glycolysis. The miR-603 directly targets *HK2* to target cellular metabolism and inhibit malignancy by acting as a tumor suppressor [78]. DNA methyltransferase 3A (DNMT3A) is a de novo methyltransferase that functions by methylating the unmethylated CpG sites. DNMT3A was overexpressed in ovarian cancer tissues when compared with normal ovary tissues. The overexpression of DNMT3A was associated with miR-603 as DNMT3A inhibited the expression of the microRNA and promoted aerobic glycolysis, cell proliferation, migration, and invasion of ovarian cancer [79]. Lactate dehydrogenase A (LDHA) is another important enzyme that is involved in glycolysis as it catalyzes the reduction in pyruvate. LDHA was upregulated in ovarian cancer tissues when compared to normal ovarian tissues [80]. Since LDHA is a key player in the conversion of pyruvate to lactate and in the maintenance of glycolysis, targeting LDHA was shown to reduce tumor growth by suppressing glycolysis. Qiu et al. showed that suppressing LDHA could inhibit lactate production, and as a result there is decreased energy supply to ovarian tumors [81]. miR-383 is an miRNA that functions as a tumor suppressor in different types of cancers. Since it functions as a tumor suppressor, this miRNA was downregulated in most tumors, and especially in ovarian cancer [82]. Ectopic expression of miR-383 induced apoptosis and inhibited cell proliferation and migration in ovarian cancer. The miR-383 was negatively correlated with LDHA in ovarian cancer tissues and it was determined to suppress LDHA by directly targeting the 3′-UTR of *LDHA* gene. Overexpression of LDHA was determined to reverse the inhibitory effect of miR-383 in ovarian cancer [83]. A study conducted by Xintaropoulou et al. evaluated the expression of different glycolytic enzymes (GLUT1, HK2 and LDHA) and their role in promoting ovarian cancer. There was significantly higher expression of GLUT1 and HK2 in high-grade serous ovarian carcinoma (HGSOC) when compared to non-HGSOC and it was also associated with advanced stages of ovarian cancer. This study also showed that when the glycolytic pathway was inhibited by using different glycolytic inhibitors, it suppressed cell growth and proliferation in ovarian cancer [84]. Therefore, these studies prove that the glycolytic pathway is an important metabolic determinant for the survival and progression of ovarian cancer and that targeting the glycolytic pathway is a potential therapeutic strategy for the treatment of ovarian cancer.

#### Outcome of Somatic Driver Mutations in Glycolysis

As previously described, Her-2/neu overexpression has been observed in different types of ovarian cancer. The overexpression of HER2 was associated with regulating tumor growth by upregulating the mTOR pathway activity and by activating the metabolic shift towards glycolysis [85]. GLUT-1 expression was positively associated with the expression of HER2 [86]. The increased expression of HER2 was also reported to elevate the expression G6PD (Glucose 6 Phosphate Dehydrogenase) [87]. HER2 overexpression also upregulates the expression of LDH-A, and targeting LDH-A using inhibitors was reported to reduce cell proliferation and survival in HER2 overexpressing tumors [88]. *c-MYC* is another oncogene that is significantly amplified and has high copy number variations in ovarian cancer and plays an important role in deregulating glycolysis. *c-MYC* was reported to upregulate the expression of GLUT1, leading to increased uptake of glucose. *c-MYC* also upregulated other important genes that are involved in glycolysis such as *LDHA*, phosphoglucose isomerase (*PGI*), phosphofructokinase (*PFK)*, glyceraldehyde-3-phosphate dehydrogenase, phosphoglycerate kinase, and enolase as well [89,90]. *KRAS* was identified as a commonly mutated oncogene in different subtypes of ovarian cancer. *KRAS* was determined to deregulate the glycolytic pathway and its components in *KRAS*-driven cancers [91]. *KRAS* mutation was reported to trigger the overexpression of GLUT1. The presence of a *KRAS* mutation and overexpression of GLUT1 worsened the survival rate of cancer patients [92]. In order to survive glucose deprivation, the *KRAS* mutant colorectal cancer cells, were found to upregulate the expression of GLUT1 and during this glucose deprivation, the wild type cells were found to acquire new mutations in the *KRAS* gene [93]. *KRAS* mutant pancreatic cancer cells were also identified to upregulate HK2 expression as well as LDHA expression [94,95,96]. *BTAK* also played an important role in deregulating glycolytic metabolism. Aurora-A kinase induced a metabolic shift towards glycolysis and altered the expression of glucose metabolic genes such as *LDHA* and *HK2* by participating and influencing the SOX8/FOXK1 signaling axis in ovarian cancer [97]. Aurora-A kinase also regulates glycolysis by stabilizing the Myc protein [98]. Aurora-A kinase is also involved in regulating the expression of metabolic genes such as *GLUT1*, *LDHA* and *HK2* expressions. Inhibiting Aurora-A kinase downregulates the expressions of these genes [99]. Aurora-A kinase was also shown to interact directly by phosphorylating LDHB, a subunit of LDH. Phosphorylation of LDHB serine 162 significantly increased its activity in reducing pyruvate to lactate, which then efficiently promoted NAD+ regeneration, glycolytic flux, lactate production and biosynthesis with glycolytic intermediates [100]. Similarly, tumor suppressor genes also play an important role in regulating the glycolytic metabolic pathway in ovarian cancer. *TP53* mutations have been observed with a high frequency rate in ovarian cancer. *TP53* was shown to mediate metabolic activities such as glycolysis in cells under both physiological and pathological conditions. *TP53* was determined to regulate glycolysis via regulating the expression of TP53-induced glycolysis regulator (TIGAR). TIGAR downregulated glycolysis by degrading fructose-2,6-bisphosphate, an allosteric effector of the glycolytic enzyme 6-phosphofructo-kinase-1 (PFK-1). TIGAR was also involved in switching the glycolytic pathway into the pentose phosphate pathway, thereby decreasing ROS generation, and promoting glutathione production [101]. Suppressing *TP53* was shown to increase the expression of GLUT1 to promote glycolysis [102]. Mutation of *TP53* was also reported to activate the expression of HK2 and phosphoglycerate mutase (PGM). Mutant *TP53* was discovered to upregulate the expression of HK2 gene and thus increase the glycolytic state in cancer cells [103]. Normal p53 expression was shown to have an inhibitory effect on expression of PGM by mediating and inhibiting the expressions of transporters GLUT1 and GLUT4 [104,105]. *BRCA1* is another major tumor suppressor gene that is reported to mutate in ovarian cancer and was determined to regulate glycolysis. A study conducted by Chiyoda et al. showed that silencing *BRCA1* increased the rate of glycolysis in ovarian surface epithelial and fallopian tube cells. The deleterious mutations in *BRCA1* caused the increased expression of HK2 and thereby promoted glycolysis [106]. *BRCA1* was also determined to regulate GLUT1. A higher expression of GLUT1 was observed in cancer cells that were carrying germline mutations for *BRCA1* [107]. Cancer cells containing mutant *BRCA1* were also determined to suppress glycolysis by repressing the genes *GLUT1*, *HK1*, *HK2*, and *LDHA*. This study also showed that *BRCA1* in turn increased the tricarboxylic acid (TCA) cycle and oxidative phosphorylation activity [108]. The tumor suppressor *PTEN* was also involved in regulation of glycolysis. Studies have identified PI3k/AKT pathway as a key regulator of GLUT1 expression [109,110]. PTEN was discovered to physically interact with AKT and cause its dephosphorylation; as a result, there is a limited expression of GLUT1 at the plasma membrane in ovarian cancer cells [111].

### 2.2. Tricarboxylic Acid Cycle

Tricarboxylic acid (TCA) cycle, also called the citric acid cycle or Kreb’s cycle, is a series of metabolic reactions occurring in mitochondria and acting as a critical source of energy for cells in the presence of aerobic conditions. The TCA cycle is present in the core of energy metabolism and is involved in macromolecule synthesis and maintaining redox balance. The TCA cycle is responsible for the production of NADH and FADH2 that fuels the electron transport chain in mitochondria for the generation of ATP. The TCA cycle is initiated when pyruvate generated from glycolysis is oxidized into acetyl-CoA by pyruvate dehydrogenase complex. It consists of eight successive reaction steps in a cyclical manner, and at the end of the cycle, one molecule of oxaloacetate becomes regenerated. The energy produced during these reactions is conserved when three NAD+ and one FAD are reduced and by the production of one ATP or GTP. The TCA cycle was deregulated in different diseases ranging from metabolic disease such as obesity to neurodegenerative disease such as Alzheimer’s disease. The TCA cycle was deregulated in ovarian cancer as well (Figure 4). Studies showed that cancer cells rely on glutamine as a fuel instead of using the pyruvate that is generated during glycolysis. In addition, when there is impaired mitochondrial pyruvate transport, glutamine is used to regulate the TCA cycle and to meet the cells’ increased metabolic needs [112,113]. In ovarian cancer, invasiveness is correlated with glutamine dependence. Low-invasive ovarian cancer was glutamine-independent, whereas invasive ovarian cancer was dependent on glycine [114]. Sometimes, the TCA cycle is also dependent on β-oxidation of fatty acids since acetyl-CoA acts as a converging point for both TCA and fatty acid metabolism [115]. Fatty Acid Synthase (FASN) is an important enzyme that converts acetyl-CoA into saturated fatty acid. FASN was highly expressed in ovarian cancer and was associated with poor survival rate [116]. In different types of cancer, and especially in ovarian cancer, the genes encoding for the enzymes aconitase, isocitrate dehydrogenase (IDH), succinate dehydrogenase (SDH) and citrate synthase (CS) were deregulated. Alterations in the genes of TCA enzymes cause the ectopic expression of different oncometabolites. Isocitrate dehydrogenase is an enzyme coded by the *IDH1* gene. This enzyme helps in conversion of isocitrate to α—ketoglutarate by oxidative decarboxylation. Isoforms of *IDH* were identified to undergo missense mutation in different types of tumors which include grade II/III gliomas and secondary glioblastomas (GBM), chondrosarcomas, and acute myeloid leukemia [117]. In ovarian cancer, wild-type *IDH1* was upregulated TCA cycle metabolism. The upregulation caused increase in the amount of α-ketoglutarate and NADPH production and also provided increased levels of reducing equivalents to sustain lipid biosynthesis and redox homeostasis [118]. Dahl et al. identified that HGSOC utilized glucose from TCA preferentially rather than from aerobic glycolysis. They also reported that *IDH1* was upregulated in ovarian cancer and was associated with reduced progression free survival. Targeting *IDH1* modifies the histone epigenetic landscape and this was discovered to induce senescence [119]. Bcl2-like-10 (*Bcl2l10*) is a member of the Bcl-2 family of genes that plays a key role in mediating apoptosis. *Bcl2l10* was identified as a tumor suppressor gene as knocking down of the gene improved cell viability, motility, and proliferation [120]. Knocking down of *Bcl2l10* was reported to deregulate the TCA cycle as some of the components of the TCA cycle acted as a downstream target of *Bcl2l10*. Succinate dehydrogenase complex subunit D (*SDHD*) and *IDH1* were regulated by *Bcl2l10*. Knocking down *Bcl2l10* downregulated *IDH1* and *SDHD* and led to the accumulation of oncometabolites such as succinate and isocitrate, and therefore lead to the promotion and progression of ovarian cancer [121]. Succinate dehydrogenase is another enzyme that is a part of the TCA cycle and is primarily involved in the catalytic conversion of succinate to fumarate by oxidation [122]. *SDH* acts as a tumor suppressor gene and consists of six subunits that encode SDHA, SDHB, SDHC, SDHD, SDHAF1, and SDHAF2 [123]. Mutations were identified in *SDH* in different types of cancer. Amplification of *SDH* was identified with a high probability rate of occurrence in HGSOC. SDHB showed a high rate of amplification compared to the other subunits [124]. Chen et al. showed that silencing *SDHB* promoted cell proliferation, migration, and invasion, whereas SDHB overexpression suppressed cell proliferation and promoted apoptosis. Silencing of *SDHB* was shown to promote overexpression of HIF-1α, a tumor-promoting factor [125]. Another study showed that knocking down *SDHB* promoted epithelial mesenchymal transition (EMT) by increased H3K27 methylation. *SDHB* knockdown also led to altered glucose and glutamine utilization and caused mitochondrial dysfunction [126]. Citrate synthase (CS) is an enzyme involved in the TCA cycle that catalyzes the reaction between oxaloacetic acid and acetyl coenzyme A to produce citrate. CS was overexpressed in malignant ovarian tumors compared to benign tumors. Knocking down *CS* using RNAi mechanism resulted in reduced proliferation, migration, and invasion in in vitro studies using ovarian cancer cell lines [127].

#### Outcome of Somatic Driver Mutations in the TCA Cycle

Oncogenes and TSG were identified to play a critical role in the control and regulation of the TCA cycle. The oncogene *c-MYC,* which is aberrantly expressed in ovarian cancer, was observed to regulate TCA either directly or indirectly. *c-MYC* acts indirectly as a master driver of glutamine metabolism through the TCA cycle. *c-MYC* also controls the conversion of glutamine to glutamate by activating glutaminase 1 (GLS1) through transcriptional suppression of its negative regulator miR-23a/b [128,129]. The *c-MYC* acts directly on the TCA cycle by interacting with the components of the TCA cycle. For example, *c-MYC* was determined to co-express with mutant *IDH1/2* and increase the state of malignancy in MYC overexpressed cancers [130]. The *c-MYC* was reported to inactivate *SDHA*. This resulted in kick-starting a regulatory cascade in cancer cells that led to the activation of H3K4me3 and induced tumor-specific gene expression and the promotion of tumorigenesis. The inhibition of SDH-complex activity led to the accumulation of succinate in cancer cells [131]. *KRAS*, which is one of the most frequently mutated oncogenes from the RAS family, plays a pivotal role in ovarian cancer metabolism by regulating the TCA cycle. *KRAS*-driven cancer cells scavenge vital proteins that contain glutamine from the extracellular space and utilize them to fuel the TCA cycle [132]. *KRAS*-driven cancer cells also scavenge branch chain amino acids such as isoleucine, valine, and leucine and convert them into acetyl-CoA to trigger the TCA cycle [133]. A study conducted by Kerr et al. showed that the copy number variant of *KRAS* promoted glucose anaplerosis fueling the TCA cycle [134]. Tumor suppressor genes such as *TP53* also play a vital role in the regulation of the TCA cycle. Pyruvate metabolism is one of the initiating factors for activating the TCA cycle. *TP53* is involved in regulating the pyruvate metabolism. The p53 transcriptionally represses the enzyme pyruvate dehydrogenase kinase-2 (PDK2), which inhibits the activity of pyruvate dehydrogenase (PDH). Using this mechanism, *TP53* helps in advancement into the TCA cycle by the conversion of pyruvate into acetyl-CoA [135]. *TP53* suppresses the expression of MCT1 (lactate/proton symporter monocarboxylate transporter 1), which reduces the ability of the cells to regenerate NAD+ through the conversion of pyruvate to lactate. Because of this, cells that lack *TP53* generate less ATP through oxidative phosphorylation when compared to cells that expresses *TP53* [136,137]. The TCA cycle can also be fueled with the help of the amino acid glutamine via α-ketoglutarate-dependent anaplerosis. Glutaminase 2 is a mitochondrial enzyme that is involved in the hydrolysis of glutamine to glutamate. *TP53* regulates the process of glutaminolysis by binding to the P53 consensus DNA-binding elements in the promoter region of *GLS2*, the gene coding for glutaminase 2. Increased expression of GLS2 enhances mitochondrial respiration, ATP generation and the production of glutathione (GSH), an anti-oxidant [138,139]. *BRCA1*, another important tumor suppressor gene, is involved in regulating metabolism by strongly inhibiting glycolysis while activating the TCA cycle and oxidative phosphorylation. Privat et al. showed that cells expressing *BRCA1* were discovered to increase the rate of transcription of *SDHC*. The enzymes *IDH1* in the cytosol and *IDH2* in the mitochondria were inversely regulated in *BRCA1* positive cells. *IDH1* was upregulated whereas IDH2 expression was downregulated. The upregulation of IDH1 could supply NADPH for glutathione reduction to combat against cellular oxidative stress [108,140]. *SDHB* was also downregulated in *BRCA1*-silenced cancer cells [141].

### 2.3. Amino Acid Metabolism

Proteins are large macromolecules that are present in the living organisms and play a critical role in every function of the body. Proteins consist of multiple sub units of amino acids. Amino acids interact with each other by forming a peptide bond and multiple peptides form complex structures that constitute the protein. Amino acids are molecules that are derivatives of carboxylic acid groups in which the α-hydrogen becomes substituted by an amino group. These molecules play an important part in sustaining life from maintaining the metabolism to act as a catalyst for various biological reactions that maintain the metabolic activities. During catabolism of amino acids, they lose their amino groups and form the “carbon skeleton” α-keto acid form. This form becomes oxidized to form CO_2_ and H_2_O. They also play a role in providing three- and four-carbon units that can be converted into glucose by the process of gluconeogenesis. Amino acid metabolism functions by maintaining the amino acid pools producing non-essential amino acids for protein biogenesis and is also involved in conversion of glucose, lipids, and nitrogen-containing metabolites, such as purines and pyrimidines for nucleic acid biosynthesis. Amino acids are required for the activation of important metabolic pathways such as the mTORC pathway and play an important role in detoxification of ammonia by converting it to the non-toxic urea form. In addition, amino acids are required for maintaining the intracellular redox homeostasis (e.g., glutathione, an important antioxidant is metabolized from the amino acids glutamate, cysteine, and glycine) [142]. The complex network of amino acid metabolism is highly interconnected with other important metabolic pathways that include glycolysis, the TCA cycle and fatty acid metabolism [143]. Since amino acid metabolism is involved in almost all the critical functions in maintaining normal cellular homeostasis, dysregulation of amino acid metabolism can be an important mediating factor in different types of cancers. Figure 5 shows the deregulated glutamine metabolic pathway in ovarian cancer. Glutamine is a non-essential α-amino acid that has an amide group that replaces the carboxylic acid group on their carbon structure. It is a highly abundant amino acid found in the body. Even though it is a non-essential amino acid, the majority of tumors require extra-cellular glutamine for their survival. Cancer cells import glutamine with the help of various transporters, which include ASC (alanine/serine/cysteine-preferring), Na^+^-dependent transporters and the Na^+^-coupled neutral amino acid transporters. Some members of these transporter families are dysregulated in different types of cancers. ASCT2, a member of ASC transporter family, was determined to play a role in the development of gastric cancer, and inhibition of ASCT2 reduced cancer growth [144]. ASCT2 was significantly upregulated in EOC. ASCT2 was positively correlated with p-mTOR in the development and progression of ovarian cancer [145]. One of the initial steps in glutamine metabolism is the conversion of glutamine into glutamate by the process of glutaminolysis. This process is catalyzed by the enzyme glutaminase (GA). GA is an enzyme coded by two isoforms of *GLS* and *GLS2*. The GA encoded by *GLS* was reported to promote tumor cell growth as it was regulated by oncogenes, and GA encoded by *GLS2* was determined to reduce the sensitivity of the cells towards ROS-associated apoptosis with the help of glutathione-dependent antioxidant defense [146]. GLS was highly expressed in cancers and to promote tumorigenesis and cell proliferation. Knocking down *GLS* was shown to suppress proliferation and promote apoptosis in prostate cancer [147]. The rate of glutaminolysis was associated with invasiveness in ovarian cancer. The higher the invasive rate, the higher the rate of glutaminolysis [148]. Glutamate is further converted into α-ketoglutarate through oxidative deamination by the enzyme glutamate dehydrogenase (GDH). α-ketoglutarate is an important substrate that is a part of the TCA cycle. Higher levels of glutamine were determined to increase the activity of the enzymes GA and GDH by modulating the mTOR/S6 and MAPK pathways in ovarian cancer [149]. GDH has been proposed as an important marker for metastasis in ovarian cancer [150]. Glutamine helps in the synthesis of the reduced form of glutathione (GSH), an anti-oxidant, as it acts as a carbon and nitrogen donor by supplementing glutamate and mediating cysteine uptake [151,152]. GSH has multiple functions. It functions as a reducing agent and an antioxidant. It mediates the metabolism of xenobiotics and also acts as a physiological reservoir for cysteine, and also regulates Ca^2+^ homeostasis [153]. GSH is also involved in inducing cellular resistance to ionizing radiation and promotes resistance against cytotoxic drugs [154]. Platinum-based chemotherapeutic drugs such as cisplatin and carboplatin are widely used for the treatment of ovarian cancer. GSH provides resistance to cisplatin and carboplatin therapy by different mechanisms, such as by reducing drug uptake and increasing intracellular drug detoxification, increased DNA repair mechanisms, and by suppressing drug-induced oxidative stress in ovarian cancer [155,156]. GSH was determined to have complex and contrasting roles in providing resistance against cisplatin. Decreased levels of reduced GSH and enzymes involved in GSH synthesis were identified to play a key role in cisplatin resistance [157]. Pompella et al. showed that the enzyme gamma-glutamyltransferase 1 (GGT1) expression on the surface of the cell favors the cellular resupply of antioxidant glutathione (GSH) which in turn favors protection against cisplatin by the formation of glycyl-cysteine dipeptide which forms an adduct with cisplatin thereby preventing its entry into the cell and its interaction with the DNA [158,159]. Platinum-resistant ovarian cancer cells showed increased dependency on glutamine metabolism as ASCT2 and GA were significantly upregulated. There is also increased dependency of glutamine utilization through the TCA cycle in platinum-resistant ovarian cancer cells. Knocking down glutaminase sensitized the resistant cells to the platinum-based compounds [160]. Glutamine, as an amide donor, is also involved in de novo synthesis of nucleotides and plays a role in maintaining the nucleotide balance. For example, purines are involved in controlling cell proliferation in ovarian cancer [161]. Myc, an important transcription factor and an oncoprotein that is found overexpressed in ovarian cancer, was shown to regulate glutamine metabolism. Increased expression of Myc was reported to promote glutaminolysis and make it highly dependent on exogenous glutamine for the survival of cell. Cells with *c-MYC* amplification undergo apoptosis because of glutamine deprivation [129,162]. Overexpressed c-Myc was reported to upregulate GA and promote glutaminolysis [163]. The above studies indicate that glutamine is equally important as glucose for cell survival and proliferation and balancing the cells’ bioenergetic needs. Serine is another important amino acid that helps in ovarian cancer progression and proliferation (Figure 6). Extra-cellular serine is required for the cancer cell progression since serine deprivation affects the tumor growth and cancer cell proliferation [164,165]. Serine is transported into the cell by Na^+^-dependent transporter ASCT1 (SLC1A4). ASCT1 is upregulated in different cancer types [166,167]. ASCT1 was also expressed in ovarian cancer and its expression was positively associated with the expression of L-type amino acid transporter 1 (LAT1), another amino acid transporter [168]. Serine metabolism is also a key player in promoting ovarian cancer [169]. Serine can be synthesized intracellularly with the help of glucose through a de novo serine synthesis pathway. The pathway is initiated when 3-phosphoglycerate (3-PG), the intermediate metabolite of glycolysis, becomes converted to 3-phosphohydroxypyruvate (3-PH). This reaction is catalyzed by 3-phosphoglycerate dehydrogenation (PHGDH). 3-PH then obtains an amino group from glutamate and forms 3-phosphoserine (3-PS) with the help of phosphoserine transaminase (PSAT) followed by dephosphorylation through serine phosphatase (PSPH) to produce serine. PHGDH, PSAT and PSPH were overexpressed in various cancers [170,171,172]. *PHGDH* was significantly upregulated at the protein level in ovarian cancer and involved in invasiveness of the cells as well as provision of resistance to platinum-based drugs [173]. Suppressing *PHGDH* was reported to inhibit proliferation, migration, and invasion, and increase cellular ROS levels in epithelial ovarian cancer [174]. PSAT1 was overexpressed in EOC. Higher PSAT1 expression indicated poor survival rate in the EOC patients and the expression was associated with increased GSH (reduced glutathione)/GSSG (oxidized glutathione) ratio and reduced ROS levels. This showed that PSAT1 promotes cancer growth by regulating the oxidation–reduction balance [175]. Glycine is an amino acid of lowest molecular weight which carries a hydrogen atom as a side-chain. Glycine is a non-essential amino acid which forms a building block for proteins as well as in different metabolic pathways such as glutathione metabolism and in regulating one-carbon metabolism [176]. Glycine is synthesized when Serine hydroxymethyl transferase (SHMT1 [cytoplasm]; SHMT2 [mitochondria]) catalyzes the transfer of the beta carbon of serine to tetrahydrofolate (THF). During this reaction, glycine is formed along with 5,10-methylene-THF, which is involved in nucleotide biosynthesis. SHMT1 was overexpressed in HGSOC and was necessary for tumor growth and cell migration [177]. SHMT2 was also overexpressed in different cancers and especially in ovarian cancer [178]. *c-MYC* which is overexpressed in ovarian cancer efficiently targets SHMT [179]. Arginine is a positively charged semi-essential amino acid and it acts as a precursor for amino acids, nitric oxide, polyamines, and creatine. Arginine is taken up into the cells by CAT-1 (SLC7A1), which is a cationic amino acid transporter (CAT). Increased expression of CAT-1 was observed in tumors that are highly L-arginine-dependent [180,181]. CAT-1 is overexpressed in EOC and it is involved in transporting phenylalanine and arginine. The expression of CAT-1 is associated with poorer survival rate in EOC patients and it was shown to promote proliferation and migration in EOC [182]. Arginine is synthesized de novo by the urea cycle. The urea cycle is a detoxification process in which toxic ammonia is converted into non-toxic urea. The substrate carbamoyl phosphate which is produced by the enzyme carbamoyl phosphate synthetase 1 (CPS1) along with ornithine is used to produce citrulline. Then, argininosuccinate synthetase 1 (ASS1), the rate-limiting enzyme involved in arginine synthesis, catalyzes the conversion of citrulline and aspartate into argininosuccinate. The enzyme argininosuccinate lyase (ASL) then cleaves argininosuccinate into arginine and fumarate. There is a relatively strong cross-talk between the enzymes involved in the urea cycle and other metabolic pathways, which promotes tumorigenesis in different cell types [183]. A reduced expression of ASS1 enzyme was observed in different types of cancers such as melanoma and hepatocellular cancers [184,185,186]. Contrastingly, higher expression of ASS1 was observed in primary high-grade and low-grade serous ovarian carcinomas and there was relatively increased expression in recurrent tumors, whereas in non-serous subtypes of ovarian cancer, a decreased expression of ASS1 was reported [187]. Silencing *ASS1* through epigenetic modification imparted ovarian cancer cells with resistance to platinum-based drugs resulting in treatment failure and clinical relapse in ovarian cancer [188]. NO is produced by the conversion of arginine catalyzed by the enzyme nitric oxide synthetase (NOS). The ASS and ASL complexes are involved in NO production along with NOS [189]. An isoform of NOS enzyme, NOS1, was upregulated in ovarian cancer. The upregulation of NOS1 promoted proliferation and invasion, and the cells with high NOS1 expression showed resistance to anti-cancer drugs [190]. The branched-chain amino acids (BCAA) such as leucine, isoleucine, and valine are group of essential amino acids that are obtained via dietary intake and scavenged by protein recycling. BCAA is transported into the cells with the help of LAT-1 (SLC7A5), a member of the SLC7 family, which consists of Na^+^- and pH-independent L-type amino acid transporters (system L/antiporter). LAT-1 is highly expressed in cancers such as breast and lung [191,192]. LAT1 is highly expressed in ovarian cancer. This transporter plays a role in cell proliferation and invasion in ovarian cancer [193]. With the help of certain highly reversible enzymes, the intracellular BCAAs are catabolized to provide nitrogen and carbon groups for various biological needs such as energy production and cell signaling [194]. The BCAA is broken down into the specific forms of branched chain keto acids (BCKA) and the branched chain amino acid transaminase 1 and 2 (BCAT1/2) simultaneously in order to produce glutamine to transfer the nitrogen to α-ketoglutarate (αKG). The enzyme-branched chain alpha-keto acid dehydrogenase complex (BCKDH) metabolizes the BCKAs in order to generate branched chain acyl-CoA (R-CoA), which then becomes further processed into the key intermediates of the TCA cycle (acetyl-CoA or succinyl-CoA (from isoleucine and valine)) [195]. BCAT enzymes comprise two isoforms: BCAT1 (cytosolic) and BCAT2 (mitochondrial). While BCAT2 is expressed in most tissues, BCAT1 is predominantly expressed in the brain, ovary, and placenta [196]. The *BCAT1* gene was notably hypomethylated in low-malignant potential (LMP) and HGSOC tumors [197]. BCAT1 was also overexpressed in LMP and HGSOC tumors. The expression of BCAT1 was associated with ovarian cancer progression as it induced cell proliferation, migration, and invasion, and inhibited cell cycle progression [198]. Branched-chain α-keto acid dehydrogenase kinase (BCKDK) acts as a negative regulator of BCAA catabolism by phosphorylating and inactivating BCKDH. A study conducted by Li et al. showed that BCKDK was highly expressed in patients with advanced pathological grade ovarian cancer. This ectopic expression of BCKDK promoted the proliferation and migration of OC cells [199].

#### Outcome of Somatic Driver Mutations in Amino Acid Metabolism

Regulation of amino acid metabolism is an important task for both oncogenes and tumor suppressor genes. Cancer cells that were positive for the oncogene *HER2* displayed higher glutamine metabolic activity compared to the non-*HER2* positive cancer cells [200]. The proteins which were associated with glutamine metabolism, such as GLS-1 and the transporter ASCT2, were highly expressed in *HER2* positive cancer cells [201]. There was also an increased expression of glycine/serine-metabolism-associated proteins such as PHGDH, PSPH and SHMT in *HER2* positive cells [202]. The oncogene *c-MYC* also plays a relatively important role in the regulation of glutamine metabolism. The MYC-driven proliferative cells exhibited a glutamine-addicted phenotype [203]. Glutamine synthetase (GS) which is a downstream target of Wnt/β-catenin pathway (an important regulator of *MYC*) was reported to interact with *c-MYC* and this induce the expression of GS through promoter demethylation [204,205]. The expression of amino acid transporters such as ASCT2 and SNAT5 was dependent on the expression of *c-MYC*. c-Myc increased both ASCT2 and SNAT5 transcriptional rate by binding in each of their promoters [129,206,207]. LAT1 and LAT2 are two amino acid transporters with a high affinity for glutamine, tyrosine, and all essential amino acids except lysine. *c-MYC* was shown to regulate LAT1 by transcriptional activation and binding to the promoter region of LAT1 in cancer cells [208]. LAT2 mRNA was enriched when MYC was expressed [206]. *c-MYC* was reported to regulate glutaminolysis by upregulating the expression of GLS1. c-Myc suppressed the micro RNA, miR-23a/b, which repressed GLS1 translation by binding to the 3′UTR of the GLS1 transcripts [163]. *c-MYC* also regulateed PSAT1 by binding to E-box sequences near the transcription start site of the PSAT1 promoter. *c-MYC*-stimulated serine biosynthesis pathway activation led to elevated levels of GSH [209]. Another important oncogene, *KRAS*, also played a major role in the regulation of amino acid metabolism. A study by Kandasamy et al. showed that *KRAS* mutant cancer cells upregulated the amino acid transporters LAT1 (SLC7A5), SNAT2 (SLC38A2), and ASCT2 (SLC1A5) [210]. *KRAS* also rewired the cells’ glutamine metabolism to increase NADPH production. Glutamine-derived aspartate should be transported into the cytosol to generate metabolic precursors for the production of NADPH; it is mainly transported by mitochondrial uncoupling protein 2 (UCP2). The UCP2-silenced *KRAS* mutant cell lines displayed decreased glutaminolysis, reduced NADPH/NADP+ and glutathione/glutathione disulfide ratios, and higher levels of ROS when compared to the wild-type counterparts [211]. The tumor suppressor gene *TP53* was also identified as a key player in mediating the amino acid metabolism. PHGDH was identified as a direct target for *TP53*. Recently, it was demonstrated that *TP53* suppresses the expression of PHGDH, thus inhibiting serine biosynthesis [212]. *TP53* was reported to upregulate the expression of proline oxidase, an enzyme involved in proline catabolism, in response to genotoxic damage, therefore regulating the balance between proline and glutamate and their derivate alpha-ketoglutarate [213]. Lowman et al. demonstrated that *TP53* significantly induced cationic amino acid transporter-3 (CAT-3) or SLC7A3, an arginine transporter. During glutamine deprivation, *TP53* activated CAT-3 and hence there was an increased influx of arginine into the cells [214]. *TP53* also induced the expression of GLS2, during stressed and nonstressed conditions in cancer cells. This, in turn, enhanced the rate of mitochondrial respiration and ATP generation and, furthermore, increased the levels of GSH and decreased ROS levels in these cells [138]. The tumor suppressor gene *PTEN* was associated with ASCT2 expression. The deletion of *ASCT2* in the presence of *PTEN* mutation resulted in metabolic stress and activation of apoptosis in cancer cells [215]. Knockdown of *PTEN* was reported to enhance BCAA catabolism in cancer cells [216].

### 2.4. Fatty Acid Metabolism

Lipids are a heterogeneous group of organic molecules that are insoluble in polar solvents and soluble in non-polar organic solvents with a diverse range of biological functions. Lipids function as an important source of energy, a key component of cell membranes, and they also participate in signaling processes. Lipids are polymers consisting of units of fatty acids (FAs). FAs form the main building blocks for several lipid species that include phospholipids, sphingolipids, and triglycerides [217]. They are composed of a carboxylic acid group with a hydrocarbon chain of varying carbon lengths and degrees of desaturation. FAs maintain the cellular lipid homeostasis as well as regulate various biochemical processes. The process of β-oxidation of FAs derives a humongous amount of energy and functions as central energy yielding pathway for various biological processes. FAs are either obtained through direct uptake from the surrounding microenvironment or they can be synthesized de novo by using nutrients such as glucose or glutamine as substrates. Cancer cells are characterized by different alterations such as alterations in transport of FAs, lipid biogenesis, lipid storage, and β-oxidation. Figure 7 shows the deregulated pathway of fatty acid metabolism in ovarian cancer. Cells import exogenous FAs with the help of different membrane transporters that facilitate efficient transport across the plasma membrane. The most well-characterized transporters include CD36 (fatty acid translocase), the members of solute carrier protein family 27 (SLC27), also known as fatty acid transport protein family (FATPs), and plasma membrane fatty acid-binding proteins (FABPpm) [218]. CD36 is a membrane glycoprotein expressed on the surface of cells. It functions by binding with the FAs and facilitates their transport across the membrane for lipid biogenesis and other processes. CD36 was upregulated in different cancer types, which includes breast cancer, acute myeloid leukemia, gastric cancer, and prostate cancer [219]. Ovarian cancer also exhibited FA uptake with the help of the CD36 transporter. A study conducted by Ladanyi et al. showed that ovarian cancer cells that were co-cultured with primary human omental adipocytes expressed increased levels of CD36 in the plasma membrane, facilitating exogenous FA uptake. Inhibiting CD36 showed the reduction in intracellular ROS levels in ovarian cancer [220]. FABP4, a transporter involved in direct transfer of lipids between adipocytes and ovarian cells, was upregulated in metastatic tumor sites. FABP4 was sufficient for diminishing the metastatic potential of HGSOC cells [221]. When there are no exogenous lipids available for import, the cells take another direction to maintain the lipid-energy homeostasis. Cells activate the pathway of lipogenesis, in which lipids are synthesized inside the cells with the help of other metabolites, in order to survive the period of starvation and energy shortage. Citrate, which is a by-product of the TCA cycle and glutamine metabolism, becomes exported into the cytoplasm from the mitochondria and acts as a precursor for lipogenesis. This cytosolic citrate is cleaved by ATP-citrate lyase (ACLY) into acetyl-CoA. The enzyme acetyl-CoA carboxylase (ACC) then converts this metabolite into malonyl-CoA. Malonyl-CoA, together with acetyl-CoA, becomes condensed into saturated fatty acids of various lengths by FASN. These fatty acids undergo further elongation with the help of fatty acid elongases. These generated fatty acids are saturated, comprising single carbon–carbon bonds. The two major amino acids generated by FASN and fatty acid elongases, palmitic acid and stearic acid, are then converted into palmitoleic and oleic acid with the help of Stearoyl CoA desaturases (SCD), SCD1 and SCD5 enzymes. Palmitoleic and oleic acids are further reduced into polyunsaturated fatty acids (PUFA) with the help of fatty acid desaturases [222,223,224]. The de novo synthesized fatty acids are involved in various biological processes that include construction and maintenance of the cell membrane, forming molecules that are involved in cell signaling processes and biogenesis of energy-storing lipids [225]. FASN, the enzyme involved in fatty acid synthesis, was overexpressed in ovarian cancer [226]. The ectopic FASN stimulated growth in ovarian cancer cells, and the FASN levels and the lipogenic activities were reported to affect cellular lipid composition [227]. The overexpression of FASN also regulated tumor immunity by inhibiting the tumor infiltrating dendritic cells and their ability to support anti-tumor T cells [228]. Another enzyme, SCD1, was also overexpressed in ovarian cancer. SCD1 was reported to promote cancer cell proliferation, migration, metastasis, and tumor growth, and many studies reported that SCD1 played a role in maintaining the characteristics of cancer stem cells in ovarian cancer [229]. Inhibiting SCD1 in ovarian cancer promoted cell death by the processes of apoptosis and ferroptosis [230]. Acyl-CoA 6-desaturase (FADS2) along with SCD1 was also aberrantly upregulated in ascites-derived ovarian cancer cells. The increased SCD1/FADS2 levels tend to elevate the levels of PUFAs. SCD1/FADS2 maintained the ROS levels, as inhibiting them disrupted the cellular/mitochondrial redox balance by downregulating lipid hydro peroxidase (GPX4) and the GSH/GSSG ratio [231].

#### Outcome of Somatic Driver Mutations in Fatty Acid Metabolism

Fatty acid metabolic reprogramming was regulated by the major oncogenes and tumor suppressor genes that were mutated, overexpressed or underexpressed in ovarian cancer. Cancer cells that are positive for the oncogene *HER2* were determined to overexpress the FASN enzyme, the key enzyme required for fatty acid biosynthesis [232]. Suppressing the expression of FASN using FASN inhibitors downregulated HER2 mRNA and reduced the tyrosine kinase activity in ovarian cancers with HER-2 overexpression [233]. The co-expression of FASN and HER-2 tends to reduce the 5-year survival rate in patients who are diagnosed with ovarian cancer [116]. *HER2*-positive cancers also showed upregulation of SCD1 mRNA expressions [234]. A study by Jia et al. reported that FASN is a downstream effector of mTORC signaling in cells that overexpress c-Myc. Suppression of FASN by either gene silencing or by soluble inhibitors effectively suppressed the proliferation rate and induced apoptosis in the presence of high c-Myc expression [235]. MYC was reported to activate the expression of the enzymes ACLY, ACC, and SCD, which are involved in fatty acid synthesis [236,237,238]. The oncogene *KRAS* was also identified as a key regulator of fatty acid metabolism. *KRAS* activation was determined to directly control the expression of genes that are involved in β-oxidation and de novo lipogenesis. The mutant version of *KRAS* was reported to regulate the intracellular fatty acid metabolism with the help of Acyl-coenzyme A (CoA) synthetase long-chain family member 3 (ACSL3), which converts fatty acids into fatty Acyl-CoA esters, the substrates that are required for lipid synthesis and β-oxidation [239]. *KRAS* was discovered to induce lipogenesis by upregulating the expression of FASN along with other enzymes that control FA metabolism, such as ACLY and ACC [240]. *KRAS* was also reported to induce genes such as sterol regulatory binding protein (*SREBP1*) and *SCD*. *SREBP* was elevated in *KRAS*-negative tumors, whereas *SCD* was elevated in *KRAS*-positive tumors. FASN was highly up-regulated in both *KRAS*-positive and -negative tumors [241]. The tumor suppressor gene, *TP53,* was also detected to regulate fatty acid metabolism by targeting the genes that play a key role in fatty acid metabolism. TP53 negatively correlated with SREBP1 expression. *TP53* transcriptionally repressed the expression of *SREBP1* [242], whereas the mutant *TP53* binds and activates the transcriptional activity of *SREBP* [243]. *TP53* also upregulated the expression of FASN with the help of MAPK [244]. Another tumor suppressor gene, *BRCA1,* regulated lipogenesis by interacting with the phosphorylated form of acetyl coenzyme A carboxylase alpha (P-ACCA) through its tandem of BRCA1 C terminus (BRCT) domains. Downregulating *BRCA1* increased the rate of fatty acid synthesis. This is because *BRCA1* affected lipid synthesis by preventing dephosphorylation of P-ACCA [245]. Ketone bodies are compounds that are produced when fatty acids are broken down for energy. The metabolites β-hydroxybutyrate, propionate, and acetone that are related to the biosynthesis of ketone bodies were downregulated by BRCA1 [108]. The loss of *BRCA1* in ovarian cancer decreased the rate of fatty acid oxidation and increased the expression of NADPH and Myc [106]. *BRCA1* was also deetrmined to upregulate the expression of FASN at the transcriptional level [108]. The forkhead transcriptional factor, FOXO1, a downstream target of PTEN/PI3K/AKT, regulated SREBP and lipogenesis by repressing SREBP transcription [246]. The loss of *PTEN* in cancer cells resulted in enhanced de novo synthesis and β-oxidation of fatty acids [216].

**Table 2 metabolites-13-00560-t002:** Differential expression of proteins present in the major metabolic pathways that are deregulated in ovarian cancer.

Metabolic Pathway	Metabolic Proteins	Differential Expression	References
Glycolysis	GLUT1	Overexpression	[73,74]
	HK2	Overexpression	[77]
	PFK	Overexpression	[89,90]
	LDHA	Overexpression	[80,81]
Tricarboxylic acid cycle	IDH1	Overexpression	[118]
	SDHB	Reduced expression	[125,126]
	CS	Overexpression	[127]
Amino acid metabolism	LAT1	Overexpression	[168,193]
	ASCT2	Overexpression	[144,145]
	BCAT1	Overexpression	[197,198]
	GLS	Overexpression	[147,148]
	ASS1	Overexpression	[187]
	BCKDK	Overexpression	[199]
	CAT1	Overexpression	[182]
	ASCT1	Overexpression	[168]
	SHMT2	Overexpression	[178]
	PSAT1	Overexpression	[175]
	PHGDH	Overexpression	[173]
Fatty acid metabolism	CD36	Overexpression	
	FABP4	Overexpression	[221]
	ACLY	Overexpression	[236,237,238]
	ACC	Overexpression	[236,237,238]
	FASN	Overexpression	[226,227,228]
	SCD1	Overexpression	[229]
	FADS2	Overexpression	[231]

## 3. FDA-Approved Drugs Targeting Ovarian Cancer and Their Role in Cancer Metabolism

Alterations in various metabolic pathways are highly essential for cancer cell survival, and therefore this can be an effective target for various therapeutic strategies for the treatment of cancer. Recently, the U.S. Food and Drug Administration (FDA) has approved a list of drugs that can be administered and can target ovarian cancer efficiently. In the next section, we summarize the role that these drugs play in targeting cancer metabolism. Table 3 lists the drugs approved by the FDA for the treatment of ovarian cancer and their role in cancer metabolism.

### 3.1. Chemotherapeutic Drugs

Melphalan is an antineoplastic agent that is a phenylalanine derivative of nitrogen mustard. Melphalan is administered orally or parenterally and it is mainly used for the treatment of multiple myeloma. Recently, melphalan was approved by the FDA for the treatment of ovarian cancer. Melphalan functions by alkylating DNA at the N7 position of the guanine residue and induces DNA inter-strand cross-linkages, leading to the inhibition of DNA and RNA synthesis, and hence inducing cytotoxicity against both dividing and non-dividing tumor cells. Melphalan was identified as a promising therapeutic agent for the treatment of ovarian cancer patients who harbored BRCA1/2 mutations [247,248,249]. In order to tackle the detrimental effect of melphalan, the cancer cells were determined to rewire their metabolic pathway. Cancer cells underwent resistance by the upregulation of glycolytic and pentose phosphate pathway enzymes and by downregulating the TCA cycle and electron transport chain proteins [250].

Carboplatin is a platinum-based compound with a wide range of antineoplastic properties. Carboplatin possesses tumoricidal activity similar to that of its parent compound, cisplatin, but it is comparatively more stable and less toxic than cisplatin. Carboplatin is used either alone or in combination with other chemotherapeutic agents to treat ovarian cancer. Carboplatin is activated in intracellular conditions to form reactive platinum complexes and binds to the nucleophilic groups such as GC-rich sites in DNA, and therefore induces intra-strand and inter-strand DNA cross-links as well as DNA-protein cross-links, which leads to the induction of apoptosis and cell growth inhibition. Fructose 2,6 bisphosphate (F2,6BP) is an important regulator for glycolysis. PFK158 is a molecule that effectively acts as an inhibitor of F2,6BP and results in reduced glucose uptake, ATP production, and lactate release, thereby leading cells into apoptosis. PFK158 was reported to synergize with carboplatin and effectively target the chemo-resistant ovarian cancer cells by inducing autophagic flux, which led to lipophagy and resulted in the downregulation of cPLA2, which is a lipid droplet (LD)-associated protein. In vivo studies using ovarian cancer mice models also verified that PFK158 in combination with carboplatin was involved in the breakdown of LDs [251].

Cisplatin is another important platinum-based compound with antineoplastic activity. Cisplatin is used to treat highly metastasized ovarian cancer either alone or in combination with other drugs. Cisplatin forms highly reactive platinum complexes that bind to nucleophilic groups such as GC-rich sites in DNA, resulting in the formation of intra-strand and inter-strand DNA cross-links, as well as DNA–protein cross-links [252]. Cisplatin was reported to cause cytotoxicity by interacting with GSH and thereby regulating the ROS homeostasis. Cisplatin-triggered ROS directly activated the mitochondrial outer membrane permeabilization and it aggravated the DNA damage in prostate cancer cells [253,254,255,256]. Cisplatin was also determined to cause cytotoxicity by interfering the glucose metabolism. Cisplatin was discovered to redirect cancer cells from the aerobic glycolysis state to oxidative phosphorylation, which is generally not preferred by cancer cells as it activates Bax and Bak, which regulates cellular apoptosis [257]. Cisplatin was also determined to downregulate the genes such as *HK2* and *PDK* that are involved in the aerobic glycolysis [258].

Cyclophosphamide is a novel synthetic alkylating agent that is chemically related to the nitrogen mustards and has antineoplastic and immunosuppressive activities. Cyclophosphamide is used as a chemotherapeutic agent for different malignancies such as acute lymphoblastic leukemia, acute myeloid leukemia, breast cancer, and ovarian cancer. Cyclophosphamide functions by being converted into the active metabolites aldophosphamide and phosphoramide mustard, which then binds to the DNA, resulting in the inhibition of DNA replication and thereby initiating cell death [259,260]. Cyclophosphamide was determined to kill cancer cells by inducing ferroptosis by increasing the levels of ROS and intracellular iron, and by decreasing the levels of GSH. Cyclophosphamide treatment was also determined to upregulate the expression of nuclear factor E2-related factor 2 (NRF2) and heme oxygenase-1 (HMOX-1) genes [261].

Doxorubicin is an anthracycline compound with anti-cancer activity. The hydrochloride salt of doxorubicin is extensively used as a chemotherapeutic agent for the treatment of various types of cancers including ovarian cancer. Doxorubicin acts by intercalating between the base pairs in the DNA helix, thereby preventing DNA replication and eventually inhibiting protein synthesis. Doxorubicin inhibits the enzyme topoisomerase II which then results in the formation of an increased and stabilized cleavable enzyme-DNA linked complex during DNA replication, which subsequently prevents the ligation of the nucleotide strand after the breakage of the double strand. Doxorubicin also generates free radicals, inducing cytotoxicity in cancer cells [262]. Doxorubicin was discovered to stimulate fatty acid oxidation and inhibit de novo lipogenesis at doses that do not induce apoptosis or changes in cell viability [263]. Doxorubicin was also reported to induce excess production of ROS in the mitochondria with the help of NADPH oxidase enzyme [264].

Gemcitabine hydrochloride is the hydrochloride salt of an analog of the nucleoside deoxycytidine which is an antimetabolite with antineoplastic function. Gemcitabine is converted intracellularly and forms the active metabolites difluorodeoxycytidine di- and triphosphate (dFdCDP, dFdCTP). dFdCDP functions by inhibiting ribonucleotide reductase, thereby decreasing the deoxynucleotide pool available for DNA synthesis, whereas dFdCTP is directly incorporated into DNA, resulting in DNA strand termination and apoptosis [265]. Cancer cells were determined to counteract gemcitabine by altering their lipid metabolism. Cancer cells were discovered to promote resistance by overexpressing the FASN enzyme when they were treated with gemcitabine [266]. Cancer cells also underwent resistance by rewiring their glucose metabolism and by enhancing the pyrimidine synthesis pathway. This study also observed a decrease in the levels of glycine and increased levels of glutathione in gemcitabine-resistant cells in order to combat the ROS generated by gemcitabine [267].

Paclitaxel is a Taxol-based compound that has anti-cancer activity. Paclitaxel functions by binding to tubulin and by inhibiting the disassembly of microtubules resulting in the inhibition of cell division. Paclitaxel also blocks the function of the anti-apoptotic protein Bcl-2 and induces apoptosis [268]. Even though paclitaxel is a microtubule inhibitor, it was discovered to affect the levels of regulation of glycolysis in cancer cells. The cancer cells which were treated with paclitaxel showed a decrease in the levels of glucose 1,6-bisphosphate and fructose 1,6-bisphosphate, the two allosteric signaling molecules in glycolysis. As a result, a decrease in ATP levels and cell viability was observed [269].

Topotecan hydrochloride is a hydrochloride salt of a semisynthetic derivative of camptothecin (a topoisomerase inhibitor). Topotecan stabilizes topoisomerase I-DNA covalent complexes thus inhibiting the relegation of topoisomerase I-mediated single-strand DNA breaks. This produces deadly double-strand DNA breaks when these complexes are encountered by the DNA replication machinery [270]. Topotecan hydrochloride showed anti-tumor effects by altering aerobic glycolysis. Topotecan in combination with DT-13, a plant-derived molecule with anti-cancer properties, was determined to suppress aerobic glycolysis by inhibiting the activities of enzymes involved in glycolytic regulation. This combination treatment degraded the epidermal growth factor receptor and inhibited the activity of HK2, eventually impeding aerobic glycolysis [271].

### 3.2. Monoclonal Antibodies

Bevacizumab is a recombinant humanized monoclonal IgG1 antibody that effectively targets vascular endothelial growth factor-A (VEGF-A), a pro-angiogenic cytokine, thereby preventing angiogenesis [272]. Cancer cells treated with bevacizumab were determined to undergo alterations in lipid metabolism. Curtarello et al. showed that when ovarian cancer xenografts are treated with bevacizumab, alterations of the tumor lipidomic profile occur, including increased levels of triacylglycerols and reduced saturation of lipid chains. Upregulation of pathways that are involved in lipid metabolism was observed during transcriptome analysis. Increased accumulation of lipid droplets was observed in tumors [273].

Mirvetuximab soravtansine-gynx is another monoclonal antibody that was recently approved by the FDA for the treatment of ovarian cancer. Mirvetuximab soravtansine-gynx is an immunoconjugate consisting of the humanized monoclonal antibody M9346A against folate receptor 1 (FOLR1) which has a potential antineoplastic activity. The M9346A antibody is conjugated to the cytotoxic maytansinoid DM4 with the help of the disulfide-containing cleavable linker sulfo-SPDB. When the antibody–antigen interaction takes place and internalization occurs, the immunoconjugate releases DM4, which binds to the tubulin and disrupts microtubule assembly/disassembly dynamics, thus inhibiting cell division [274]. Folate metabolism was involved with amino acid metabolism since folate coenzymes are required for the metabolism of several important amino acids, such as methionine, cysteine, serine, glycine, and histidine. Binding and blocking the receptors in FOLR1-expressing cells can deregulate the amino acid metabolic pathway [275].

### 3.3. Small Molecule Inhibitors

Olaparib is a small molecule that acts as an inhibitor of the nuclear enzyme poly (ADP-ribose) polymerase (PARP). Olaparib selectively binds and inhibits PARP, thereby preventing PARP-mediated repair of single-strand DNA breaks. BRCA1/2 mutations are the generally observed major mutations in ovarian cancer. Studies have shown that inhibition of PARP activity induced synthetic lethality in BRCA1/2 mutant tumors. Therefore, drugs such as Olaparib and niraparib were approved for the treatment of BRCA mutated ovarian cancer [276]. Olaparib can be used in ovarian cancer patients who are having only partial response to chemotherapeutic agents as it enhances the cytotoxicity of DNA-damaging agents and can reverse tumor cell chemoresistance by inhibiting PARP [277]. Berardi et al. showed that metabolic alterations were observed in cancer cells post-Olaparib exposure. This study observed metabolic reprogramming of amino acids (glutamine, glutamate, aspartate, alanine, arginine, and proline) as well as lipid metabolism (butanoate metabolism) [278].

Niraparib tosylate monohydrate is a tosylate salt form of niraparib which is a PARP-1 and PARP-2 inhibitor and has an effective and powerful antineoplastic activity against ovarian cancer. Niraparib causes genomic instability by blocking PARP-1 and -2-mediated DNA repair, thereby leading to the accumulation of DNA strand breaks and resulting in apoptosis [279]. Cancer cells were discovered to adapt against niraparib by rewiring glucose metabolism by impacting glycogen synthase. The GSK3-induced phosphorylation of the enzyme glycogen synthase caused reduced synthase activity, inferring that tumor cells that are adapted to niraparib will have lesser stored glucose (glycogen), and any glucose taken up by the adapted cell will be mostly utilized for metabolic processes rather than be stored [280]. PARP inhibitors such as Olaparib and Niraparib were determined to induce a metabolic shift from anaerobic glycolysis to tricarboxylic acid cycle resulting in increased ATP synthesis. PARP inhibitors were also reported to limit PARP-1-induced depletion of intracellular NAD+, and, as a result, they promote alternative mechanisms of cell death caused by ROS induction and excess ATP production by maintaining the mitochondrial bioenergetics [281,282].

**Table 3 metabolites-13-00560-t003:** The list of FDA-approved drugs for the treatment of ovarian cancer and their role in cancer metabolism.

Class of the Drug	FDA-Approved Drug	Role of the Drug in Cancer Metabolism	References
Chemotherapeutic drugs	Melphalan	Melphalan efficiently targeted the DNA repair mechanisms in ovarian cancer patients with BRCA1/2 mutations.	[247,248,249]
	Carboplatin	Carboplatin in combination with PFK158 promoted lipophagy in chemoresistant cells.	[251]
	Cisplatin	Cisplatin interacted with GSH and regulated ROS homeostasis. Cisplatin redirected the cancer cells from the aerobic glycolysis to oxidative phosphorylation. Cisplatin downregulated HK2 and PDK.	[253,254,255,256,257,258]
	Cyclophosphamide	Cyclophosphamide was identified to trigger ferroptosis by increasing the ROS and intracellular iron levels and by decreasing GSH levels.	[261]
	Doxorubicin	Doxorubicin stimulated fatty acid oxidation and inhibit de novo lipogenesis by the activation of p53. Doxorubicin induced ROS in the mitochondria, with the help of NADPH oxidase enzyme.	[263,264]
	Gemcitabine hydrochloride	The overexpression of the FASN enzyme promoted resistance in cancer cells that were treated with gemcitabine.	[266]
	Paclitaxel	Paclitaxel decreased the levels of glucose-1,6-bisphosphate and fructose-1,6-bisphosphate and caused the detachment of phosphofructokinase from the cytoskeleton of cancer cells.	[269]
	Topotecan hydrochloride	Topotecan in combination with DT-13 inhibited HK2 activity which in turn suppressed aerobic glycolysis.	[271]
Monoclonal antibodies	Bevacizumab	Bevacizumab induced lipid metabolic rewiring and upregulated pathways that are involved in lipid metabolism in ovarian cancer.	[273]
	Mirvetuximab soravtansine-gynx	Mirvetuximab deregulated folate metabolism by binding and blocking FOLR1 receptor.	[275]
Small molecule inhibitors	Olaparib	Enacted metabolic reprogramming of glutamine-derived amino acids and lipid metabolism in Olaparib-treated cancer cells. Olaparib caused metabolic shift from anaerobic glycolysis to tricarboxylic acid cycle which was induced resulting in increased ATP synthesis.	[278,281,282]
	Niraparib tosylate	Niraparib-adapted tumor cells had lesser levels of stored glucose as GSK3 phosphorylates the enzyme glycogen synthase. Niraparib induced a metabolic shift to tri-carboxylic acid cycle from anaerobic glycolysis which resulted in increased ATP synthesis.	[280,281,282]

## 4. Conclusions and Future Perspective

Several studies have been conducted and significant progress has been achieved in the past decades to understand the causes and the consequences of metabolic reprogramming in cancer. Recent studies have identified that metabolic reprogramming is the result of genetic alterations of various oncogenes and tumor suppressor genes. Even though the exact molecular mechanism of how precisely these genes are involved in altering cellular metabolism is still unknown, it has been shown that these genes indirectly play a role in the production, transport, and elimination of metabolites that are derived from different metabolic pathways. As a result, the alteration in the levels of the metabolites can have a drastic effect on cell signaling, epigenetics, and gene expression that are required for their survival. Several cancer models were used to study and dissect the metabolic circuits, but in the case of ovarian cancer, the mechanisms involved in regulating the metabolic pathways and the role that the oncogenes and tumor suppressor genes play in mediating the metabolic pathways remain understudied. A recent study showed that ovarian cancer cells post ROS exposure did not show any significant alterations in the overall consumption of metabolic substrates across different metabolic pathways. The consumption of metabolic substrates remained more or less similar to that of the cells that were not exposed to ROS [283]. This proves that identifying a proper therapeutic agent that effectively targets ovarian cancer is a difficult task. Recently, FDA has approved a list of drugs for the treatment of ovarian cancer, and some of the promising drugs such as metformin which effectively targets glucose metabolism are still in the clinical trials [284,285]. The role of the approved drugs in targeting the metabolism in ovarian cancer is yet to be understood clearly and developing new drugs that target the ovarian cancer metabolism can be a promising treatment modality. Therefore, directly targeting the metabolic pathways can be an important strategy for designing drugs for personalized therapies for patients with ovarian cancer.

## Figures and Tables

**Figure 1 metabolites-13-00560-f001:**
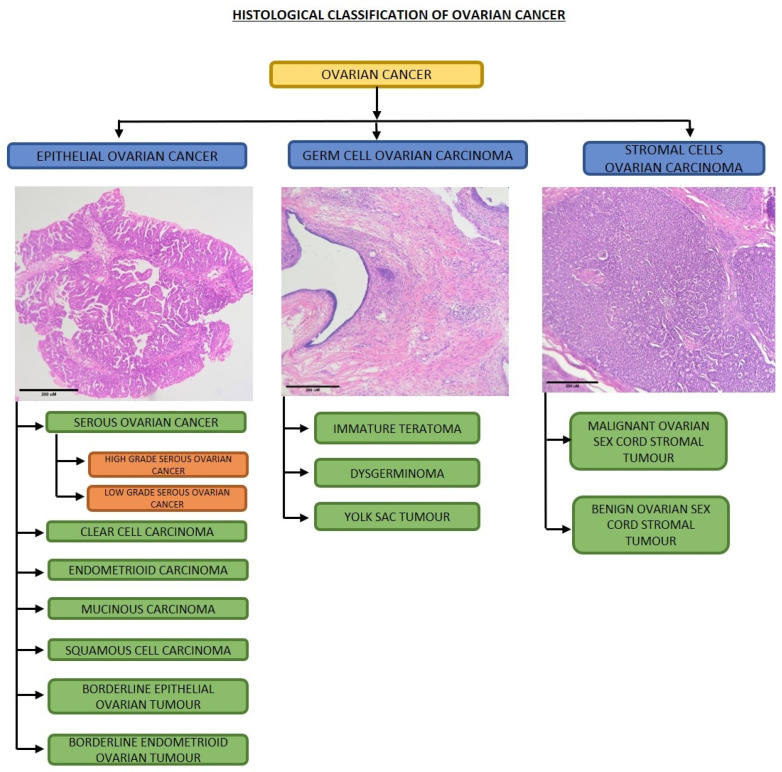
Schematic representation of histological classification of ovarian cancer.

**Figure 2 metabolites-13-00560-f002:**
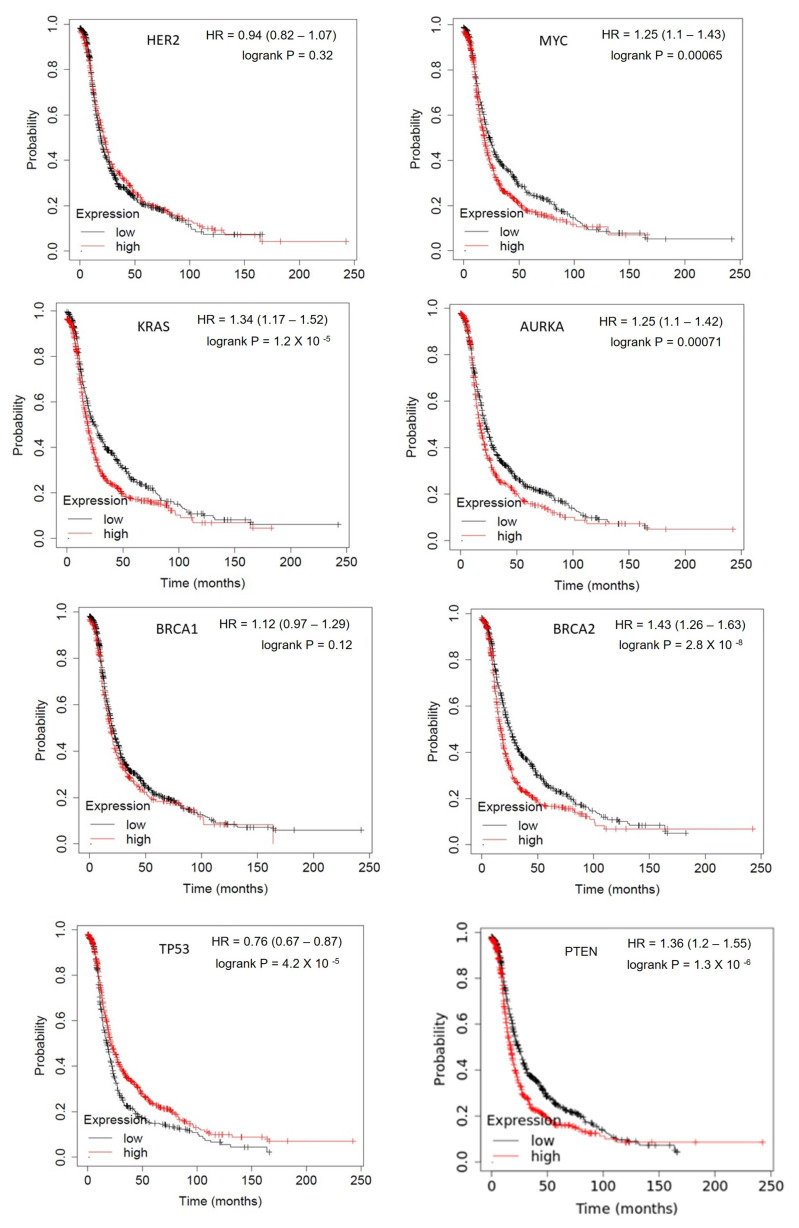
The overall survival of patients (*n* = 1435) from the TCGA database harboring mutation in the oncogenes and tumor suppressor genes.

**Figure 3 metabolites-13-00560-f003:**
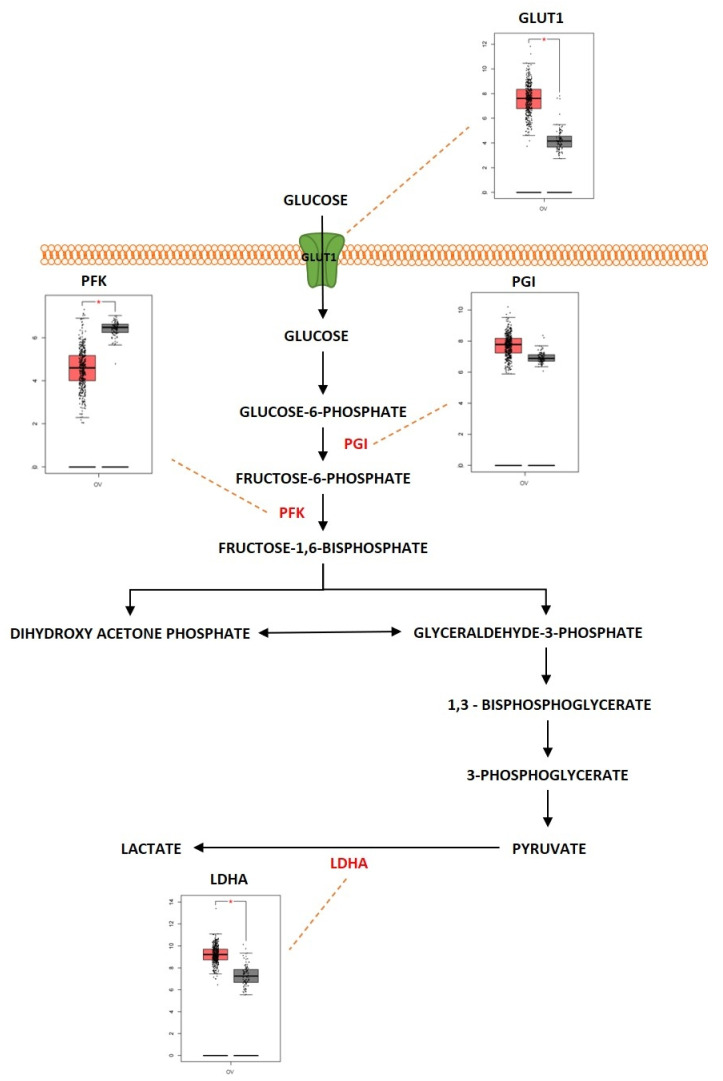
Differentially expressed proteins of the glycolysis pathway in ovarian cancer. Schematic presentation of glycolytic pathway in ovarian cancer and highlight of key differentially regulated genes (GLUT1; overexpressed, PFK; underexpressed, PGI; overexpressed, LDHA; overexpressed). Boxplot derived from TGCA expression datasets with tumor (red) and control (gray). * Represents statistical significance of *p* < 0.05.

**Figure 4 metabolites-13-00560-f004:**
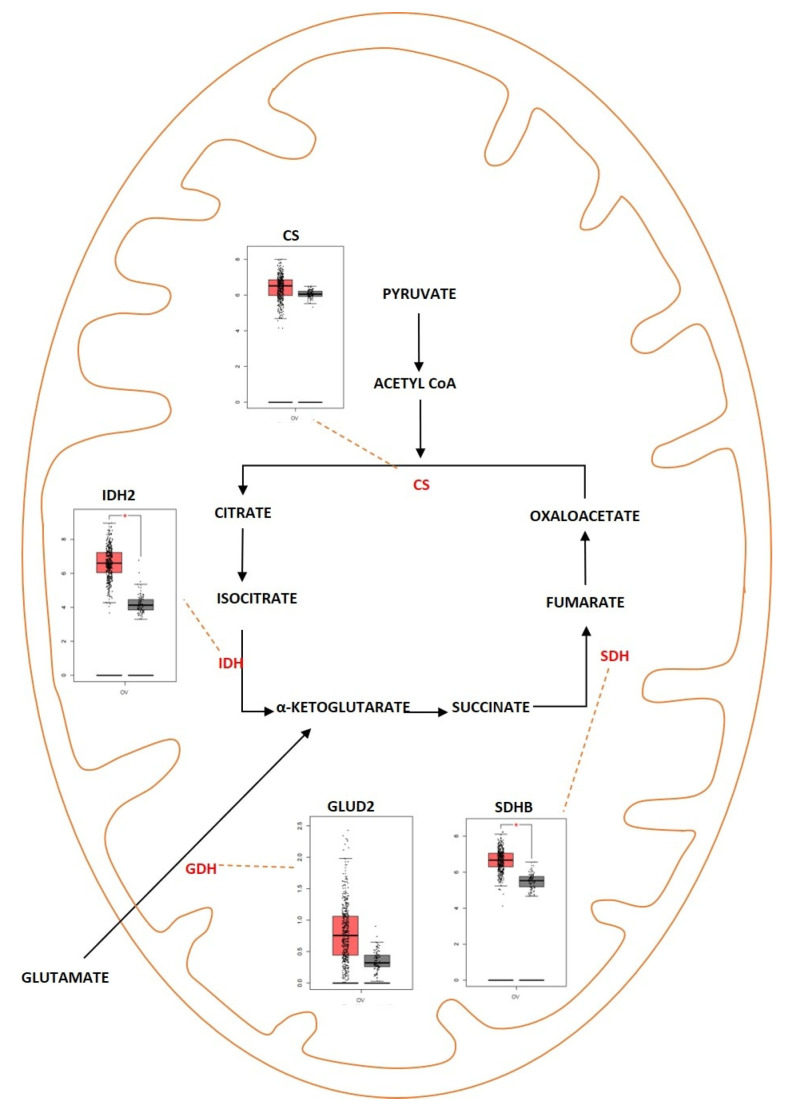
Differentially expressed proteins of the TCA cycle in ovarian cancer. Schematic representation highlighting the differentially regulated genes of the TCA cycle in ovarian cancer. (CS; overexpressed, IDH2; overexpressed, SDHB; overexpressed, GLUD2; overexpressed). Boxplot derived from TGCA expression datasets with tumor (red) and control (gray). * Represents statistical significance of *p* < 0.05.

**Figure 5 metabolites-13-00560-f005:**
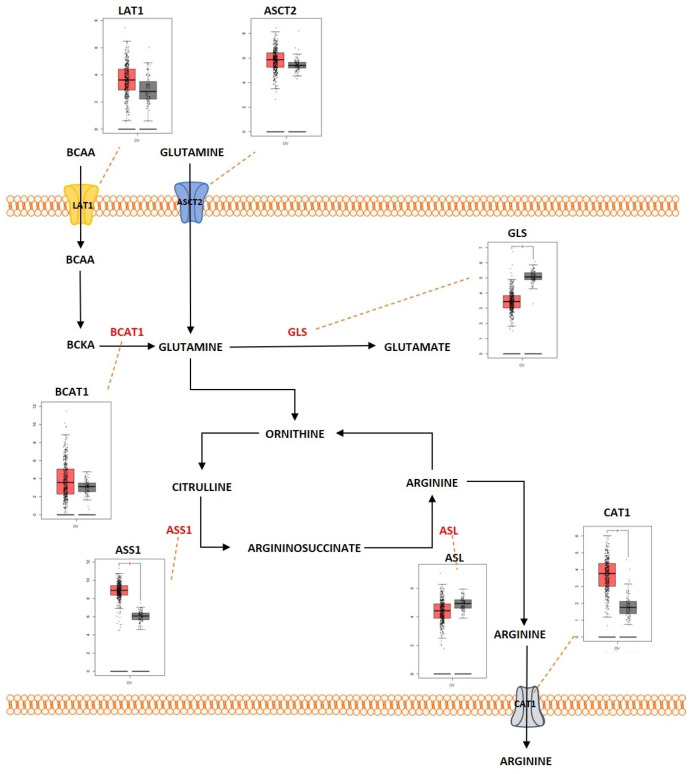
Differentially expressed proteins of the glutamine metabolic pathway in ovarian cancer. Schematic representation of the glutamine metabolic pathway in ovarian cancer that highlights the differentially regulated genes (LAT1; overexpressed, ASCT2; overexpressed, BCAT1; overexpressed, GLS; overexpressed, ASS1; overexpressed, ASL; underexpressed; CAT1; overexpressed). Boxplot derived from TGCA ex-pression datasets with tumor (red) and control (gray). * Represents statistical significance of *p* < 0.05.

**Figure 6 metabolites-13-00560-f006:**
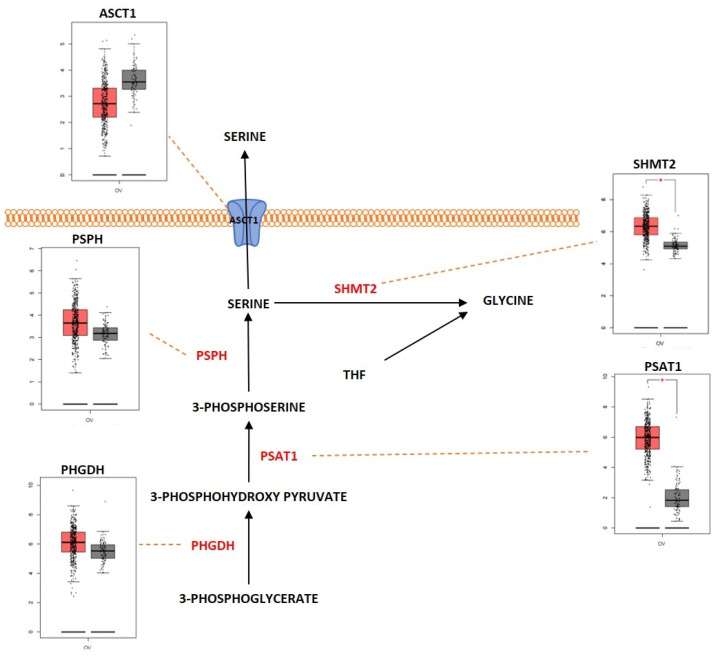
Differentially expressed proteins of the serine metabolic pathway in ovarian cancer. Schematic representation of the serine metabolic pathway that highlights the differentially regulated genes in ovarian cancer (ASCT1; underexpressed, SHMT2; overexpressed, PSPH; overexpressed, PSAT1; overexpressed, PHGDH; overexpressed). Boxplot derived from TGCA expression datasets with tumor (red) and control (gray). * Represents statistical significance of *p* < 0.05.

**Figure 7 metabolites-13-00560-f007:**
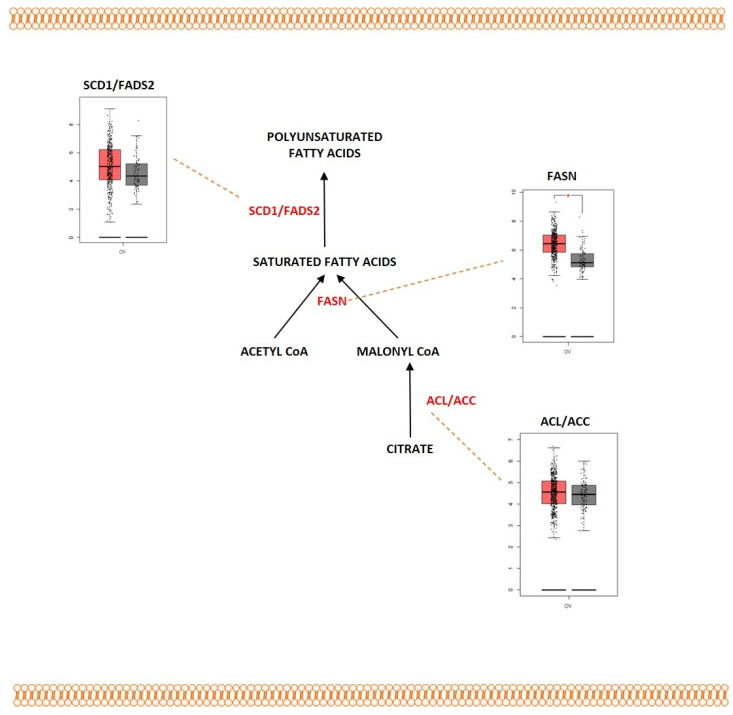
Differentially expressed proteins of the fatty acid metabolic pathway in ovarian cancer. Schematic representation of fatty acid metabolic pathway in ovarian cancer and highlight of key differentially regulated genes (SCD1/FADS2; overexpressed, FASN; underexpressed, ACL/ACC; overexpressed). Boxplot derived from TGCA expression datasets with tumor (red) and control (gray). * Represents statistical significance of *p* < 0.05.

**Table 1 metabolites-13-00560-t001:** The list of major oncogenes and tumor suppressor genes that contribute to the pathogenesis of ovarian cancer.

Class of the Gene	Frequency of Somatic Mutations	Frequency of Germline Mutations	Gene	Genetic Alterations	References
Oncogenes	0.9%		*Her-2/neu*	Amplification	[22,23,24]
			*c-MYC*	Amplification Copy Number Variation	[25,26,27]
	0.6%		*KRAS*	Mutations in codon 12 and 13	[28,29,30]
			*BTAK*	Amplification	[31,32]
Tumor Suppressor Genes	95.9%		*TP53*	Loss of Heterozygosity Driver Mutations Missense Mutations	[33,34,35,36,37,38]
	3.5%	8.2%	*BRCA1*	Germline and Somatic mutations	[36,37,38]
	3.2%	7.9%	*BRCA2*	Germline and Somatic mutations	[39,40,41]
	0.6%		*PTEN*	Driver Mutations	[39,40,41,42,43]

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
