# Peer review of "Deregulated Metabolic Pathways in Ovarian Cancer: Cause and Consequence"

_metabolites, 2023, doi:10.3390/metabo13040560_

Round 1

Reviewer 1 Report

This manuscript is well written and the logic is clear. I agree with its publication. However, there are some parts that could be complemented before publication.

General comments

(1) The figure is too small and hard to see. We recommend enlarging it.

(2)The specific mechanisms should be focused on more than simply the up- and down-regulation of molecules. This should be the focus of the revision.

Specific comments

(1)Somatic Driver Mutations 

The data related to Somatic Driver Mutations is provided in TCGA. I think it would be more appropriate to display them and sort them by frequency of occurrence of mutation c in the later section.

The relationship between BRAC and PARP should be highlighted. This is because PARP inhibitors are very promising targeted agents for ovarian cancer.

(2)Deregulated metabolic pathways in ovarian cancer

The diagram shows the metabolic pathways and the genes that are dysregulated. This approach is feasible. However, it would be more appropriate to visualize the current research on a metabolic pathway, including molecules, metabolites, and mechanisms, as mentioned in the manuscript. Figures or tables are acceptable forms.

(3)FDA approved drugs targeting ovarian cancer and their role in cancer metabolism

It would be better if the drugs could be presented again in a table format to increase readability.

Author Response

Reviewer 1

This manuscript is well written and the logic is clear. I agree with its publication. However, there are some parts that could be complemented before publication.

General comments

(1)        The figure is too small and hard to see. We recommend enlarging it.

Size of the images has been enlarged as recommended

(2)        The specific mechanisms should be focused on, more than simply the up- and down-regulation of molecules. This should be the focus of the revision.

We thank the reviewer for this comment. We have already discussed in the text the consequences of the gain/loss of function mutations. For example, in paragraph 2.1.1 we have discussed the role of her2 mutations in glycolysis and the association of GLUT1 G6PD and LDHA. This applies to other oncogenes and tumor suppressor genes. The underlying mechanism remains to be investigated to deduce druggable targets in ovarian cancer.

Specific comments

  • Somatic Driver Mutations

The data related to Somatic Driver Mutations is provided in TCGA. I think it would be more appropriate to display them and sort them by frequency of occurrence of mutation c in the later section.

The relationship between BRAC and PARP should be highlighted. This is because PARP inhibitors are very promising targeted agents for ovarian cancer.

We thank the reviewer for this comment, we have now introduced the somatic and germline frequencies in table 1. The relationship and BRAC and PARP are also discussed in paragraph 3.3.

  • Deregulated metabolic pathways in ovarian cancer

The diagram shows the metabolic pathways and the genes that are dysregulated. This approach is feasible. However, it would be more appropriate to visualize the current research on a metabolic pathway, including molecules, metabolites, and mechanisms, as mentioned in the manuscript. Figures or tables are acceptable forms.

We have now introduced Table 2 to visualize the current research in metabolic pathways for better readability to the audience

  • FDA approved drugs targeting ovarian cancer and their role in cancer metabolism

It would be better if the drugs could be presented again in a table format to increase readability.

Table 3 has been introduced summarizing FDA approved drugs in ovarian cancer metabolism

Reviewer 2 Report

The manuscript by Murali et al. titled “Deregulated Metabolic Pathways in Ovarian Cancer: Cause and Consequence” is a review article aimed to: i) highlight the genetic alterations undergone by the key oncogenes and tumor suppressor  genes responsible for the development of ovarian cancer; ii) summarize the role of these oncogenes and tumor suppressor genes and their association with a deregulated network of fatty acid, glycolysis, tricarboxylic acid and amino acid metabolism in ovarian cancers.

The topic of the manuscript is of great interest and relevance. However, the manuscript requires major revisions before acceptance for publication.

Major comments:

1.The first paragraph (under the heading 1.Introduction) is somehow incomplete. For example, the mechanisms and molecular pathways involved in the pathogenesis and carcinogenesis are different between high grade and low grade serous, clear cell and endometrioid ovarian cancer. This concept should be introduced. Moreover, the sensitivity to chemotherapy may differ according to histology, molecular profile and p53 status. Therefore, some biomarkers are known. However, there is a lack in the application of targeted therapies, especially in low grade ovarian cancer despite the peculiar molecular aberrations that have been hypothesized and found to be involved in its pathogenesis. 

2.Table 2 seems to report original data on survival. It is not clear whether these data have been elaborated by the authors or they refer to a specific manuscript. In the first case the authors should explain how they elaborated the data; in the latter case the reference should be cited.

3.Table 1. The list of genes is incomplete. In particular the gene linked to the PI-3KI_AKT_mTOR pathways are lacking, while they have been involved especially in low grade (clear cell and endometrioid) ovarian cancers.

4.Line 552-555: GSH has a complex and controversial role in mediating resistance to cisplatin. Please check also other references as PMID: 16303117, PMID: 36009263, PMID: PMID: 35669424

5.In the Figures 3,4,5,6, and 7 box plots are reported. Did they refer to original data of the authors. Did they refer to a specific publication?. In the first case the authors should explain how they elaborated the data. in the latter case they should cite the references.

6.Lines 865-872: Cisplatin has multiple mechanism of action. Notably it also acts on glycolysis (hexokinase 2 and PDK): please check and cite the ref PMID: 23876094.

7.Line 928. “immunotherapeutic drugs”: The term immunotherapeutic is confounding because these drugs are not immunotherapy, but monoclonal antibodies developed against different targets by an immunological technique. I would suggest changing the term.

8.Paragraph 3.3. The PARP inhibitor have additional relevant effects, such as those on the mitochondrial oxidative phosphorylation, PDK, HK2 and other, see the ref PMID: 31070782.

9.The paragraph “FDA approved drugs targeting ovarian cancer and their role in cancer metabolism” is long and could benefit and be clearer by adding a figure that summarize the main effect of the different drugs on the main metabolic pathways discussed in the manuscript.

10.Lines 989-990 “The FDA has recently approved a plethora 989 of drugs for various cancers, but in the case of ovarian cancer, only a few drugs were 990 approved for its treatment and most of the drugs are in pre-clinical trials” The sentence is unclear. Did you mean drug against alteration of metabolic pathways such as glycolysis etc…? In this case please cite at least the main drugs developed with the corresponding reference.

Minor comments:

1.Line 185: It could be useful to know the percentage of mutation associated with LOH, because it can influence sensitivity to chemotherapy (cisplatin).

2.Line 232: please cite the reference

Author Response

Reviewer 2

Major comments:

1.The first paragraph (under the heading 1.Introduction) is somehow incomplete. For example, the mechanisms and molecular pathways involved in the pathogenesis and carcinogenesis are different between high grade and low grade serous, clear cell and endometrioid ovarian cancer. This concept should be introduced. Moreover, the sensitivity to chemotherapy may differ according to histology, molecular profile and p53 status. Therefore, some biomarkers are known. However, there is a lack in the application of targeted therapies, especially in low grade ovarian cancer despite the peculiar molecular aberrations that have been hypothesized and found to be involved in its pathogenesis.

We thank the reviewer for the above comment. We have now made substantial changes in the introduction and discussed about the dualistic model of ovarian carcinogenesis. We have included existing biomarkers and its applications in therapies and the gap in the field to identify additional reliable markers and therapeutic targets.

2.Table 2 seems to report original data on survival. It is not clear whether these data have been elaborated by the authors or they refer to a specific manuscript. In the first case the authors should explain how they elaborated the data; in the latter case the reference should be cited.

The survival datasets in Figure 2 are elaborated from KMplotter™ that mines the TCGA datasets. We have introduced appropriate citations for the figure 2

3.Table 1. The list of genes is incomplete. In particular the gene linked to the PI-3KI_AKT_mTOR pathways are lacking, while they have been involved especially in low grade (clear cell and endometrioid) ovarian cancers.

We thank the reviewer for highlighting this. We have now also introduced PTEN as the tumor suppressor gene involved in PI3K/AKT/mTOR pathways in Table 1, paragraph 1.1.8 and 2.1.1, 2.3.1 and 2.4.1

4.Line 552-555: GSH has a complex and controversial role in mediating resistance to cisplatin. Please check also other references as PMID: 16303117, PMID: 36009263, PMID: PMID: 35669424

We thank the reviewer for highlighting the contrasting role of GSH in cisplatin resistance, we have discussed this in paragraph 2.3

5.In the Figures 3,4,5,6, and 7 box plots are reported. Did they refer to original data of the authors. Did they refer to a specific publication? In the first case the authors should explain how they elaborated the data. in the latter case they should cite the references.

We have now cited these figures with TCGA and GEPIA references

6.Lines 865-872: Cisplatin has multiple mechanism of action. Notably it also acts on glycolysis (hexokinase 2 and PDK): please check and cite the ref PMID: 23876094.

We have introduced the additional mechanisms of cisplatin influence in glycolysis in cancers in paragraph 3.1

7.Line 928. “immunotherapeutic drugs”: The term immunotherapeutic is confounding because these drugs are not immunotherapy, but monoclonal antibodies developed against different targets by an immunological technique. I would suggest changing the term.

We have replaced this with term monoclonal antibody in paragraph 3.2 and table 3

8.Paragraph 3.3. The PARP inhibitor have additional relevant effects, such as those on the mitochondrial oxidative phosphorylation, PDK, HK2 and other, see the ref PMID: 31070782.

We have discussed the mechanism of PARP inhibitors in influencing mitochondrial bioenergetics, ATP elevation and oxidative stress induced cell death in paragraph 3.3

9.The paragraph “FDA approved drugs targeting ovarian cancer and their role in cancer metabolism” is long and could benefit and be clearer by adding a figure that summarize the main effect of the different drugs on the main metabolic pathways discussed in the manuscript.

Table 3 has been now introduced for better readability and clarity of the topic.

10.Lines 989-990 “The FDA has recently approved a plethora 989 of drugs for various cancers, but in the case of ovarian cancer, only a few drugs were 990 approved for its treatment and most of the drugs are in pre-clinical trials” The sentence is unclear. Did you mean drug against alteration of metabolic pathways such as glycolysis etc…? In this case please cite at least the main drugs developed with the corresponding reference.

We thank the reviewer for this comment. We have now rephrased this section to rationalize the importance of understanding and targeting metabolic pathways in ovarian cancer

 Minor comments:

1.Line 185: It could be useful to know the percentage of mutation associated with LOH, because it can influence sensitivity to chemotherapy (cisplatin).

We have discussed the LOH in TP53 and its association with cisplatin sensitivity in ovarian cancer in paragraph 1.1.5

2.Line 232: please cite the reference

Appropriate citation has been added.

Round 2

Reviewer 1 Report

I am satisfied with the revised version.

Reviewer 2 Report

The authors have adequately addressed my comments and revised the manuscript accordingly. The revised mansucript can be accepted for publication.